

# Enhanced MODIS-derived ice physical properties within CoLM revealing bare ice-snow-albedo feedback over Greenland

Shuyang Guo[1], Yongjiu Dai[1]*, Hua Yuan[1], Hongbin Liang[1]

[1] Southern Marine Science and Engineering Guangdong Laboratory (Zhuhai), School of Atmospheric Science, Sun Yat-sen University, Zhuhai, China

Corresponding author: Yongjiu Dai (daiyj6@mail.sysu.edu.cn)

13                           Submitted to *The Cryosphere*

**Abstract**

Under global warming, the Greenland Ice Sheet (GrIS) is experiencing unprecedented mass loss. One key contributing factor is the change in snow and ice albedo, which is closely related to firn metamorphism. To investigate the impact of bare ice microstructure changes on the regional warming of the GrIS ablation zone, SNICAR-ADv4 (Snow, Ice and Aerosol Radiation model Adding-Doubling Version 4), a physically based radiative transfer model, is incorporated in the Common Land Model version 2024 (CoLM2024). It allows the land surface model to represent the ice albedo with changes in ice properties rather than using a constant ice albedo value. Quality control was conducted on the bare ice physical property dataset input into CoLM, with multiple MODIS products combined to ensure accuracy. The application of SNICAR-ADv4 reduced the overestimation of shortwave broadband albedo by 38%, with a bias of 0.053. Further sensitivity experiments indicate that the summer albedo in the bare ice region is reduced by 0.032 due to the bare ice metamorphism, producing a 2-m temperature forcing of 0.071°C, a snow cover change of -0.011, and a snow water equivalent forcing of -1.345 mm. These changes lead to increased bare ice exposure, further reducing albedo and enhancing solar radiation absorption by the surface, thereby reinforcing a feedback involving bare ice, snow, and albedo. This highlights the critical role of bare ice physical properties in amplifying melt through the bare ice-snow-albedo feedback, with stronger feedbacks expected in a fully coupled land-atmosphere model.

**Keywords**

Greenland Ice Sheet; Bare ice region; Ice albedo; Albedo feedback; MODIS; Remote sensing

**1. Introduction**

The Greenland Ice Sheet (GrIS) has been melting at a rapid pace since the 1990s, losing around 255 Gt of ice annually in 2003-2016 (Sasgen et al., 2020; Li et al., 2022; van den Broeke et al., 2017). The decreasing mass balance of the GrIS and peripheral glaciers is the

most significant cryospheric factor driving sea level rise, contributing over 25% of observed global sea level rise (Chen et al., 2017; Ryan et al., 2019). The total mass loss from GrIS consists of two components: surface runoff and frontal ablation occurring at the terminus of outlet glaciers (Cogley et al., 2011, Kochtitzky et al., 2023). Surface losses have exceeded dynamical losses in contributing to GrIS mass loss since 2000, with 55% of Greenland's total mass loss attributed to surface mass balance (SMB) and 45% to the discharge of outlet glaciers between 2000-2018 (Mouginot et al., 2019). These melting processes are driven by a combination of factors, including atmospheric warming, a reduced water retention capacity of firn due to densification, and a lower surface albedo (Hofer et al., 2017; King et al., 2020; Ryan et al., 2024).

Ice melt on the surface of the GrIS is partially regulated by the surface albedo. It serves as a fundamental parameter in controlling the absorption of insolation by the ice sheet (Box et al., 2012; Naegeli et al., 2017, Feng et al., 2024). A minor change in snow and ice surface albedo can exert a substantial effect on the energy budget of the regional surface-air system, causing significant fluctuations in the energy flux on the surface of the GrIS (Nolin and Stroeve, 1997). Surfaces with high albedo, such as fresh snow, efficiently reflect solar radiation, whereas darker areas, such as glacier ice, absorb the majority of incoming shortwave energy (Whicker-Clarke et al., 2022). Snow and ice albedo varies with the spatial distribution of snow, ice, and biotic and abiotic light absorbing constituents (LACs) and further evolves with the melting of snowpack and glacier surfaces through the spring and summer. Fluctuations in the snowline dictate the relative extent of dark bare ice versus brighter snow (Ryan et al., 2019). These directly influence GrIS surface melt through the exposure of bare ice (Antwerpen et al., 2022) and the processes that darken bare ice itself (Chevrollier et al., 2023). Dark bare ice extent closely tracks interannual variations in snowline elevation and is exposed as the snowline retreats further inland during the melt season, leading to the reduction of ice sheet albedo and the intensified melt. This positive feedback has been referred to as the "snow-albedo feedback" (Ryan et al., 2019).

In the preceding decades, polar amplification has contributed to the progressive darkening of

the GrIS and the prolongation of the melt season, both of which serve as positive feedback mechanisms that intensify surface melt (Tedesco et al., 2016). As the warming occurs over the ice surface, bare ice albedo is reduced through melt processes that darken the ice surface. Notably, these processes include exposure of dust layers, pooling of surface meltwater, increased interstitial water content, and liquid meltwater-induced growth of pigmented ice algal assemblages that inhabit the bare ice surface (Cook et al., 2020; Stibal et al., 2017; Tedstone et al., 2020; Williamson et al., 2018; Whicker-Clarke et al., 2022). Despite operating over a relatively small area of the ice sheet, it is argued that these bare ice processes have contributed substantially to an observed reduction in albedo and associated increase in melt across GrIS's ablation zone from 2000 to 2011 (Stibal et al., 2017; Tedstone et al., 2017). This category of physical and biological melt-albedo processes that darken bare ice is referred as the "bare ice-albedo feedback" (Ryan et al., 2019). However, the complex and non-linear response of regional snow and ice, particularly in ablation zones, to changes in meteorology and climate highlights the growing necessity to model these surfaces using physical principles rather than relying solely on empirical methods (Box et al., 2012). Therefore, accurately modeling the influence of snow and ice on the albedo of the GrIS becomes increasingly important to capture these dynamics effectively.

The albedo of the cryosphere varies widely depending on the solar zenith angle (SZA), atmospheric conditions, metamorphic state of the snow and ice, and impurities (He and Flanner, 2020). The Snow, Ice, and Aerosol Radiative (SNICAR) model is one of the most widely used snowpack radiative transfer models (Flanner et al., 2021). Initially, it combined the theory from Wiscombe and Warren (1980) and Warren and Wiscombe (1980) with the multi-layer two-stream solution from Toon et al. (1989) to enhance the simulation of snow albedo (Flanner et al., 2007). Updates and new features have also been incorporated within SNICAR, including eight species of LACs (Flanner et al., 2007), four snow grain shapes (He et al., 2018), black carbon-snow and dust-snow internal mixing state (Flanner et al., 2012; He et al., 2017, 2019). Dang et al. (2019) developed SNICAR-AD by substituting the tri-diagonal matrix solution solving method (Toon et al., 1989) with the delta-Eddington adding-doubling radiative method, as a result of the latter's superior computational stability

across varying solar zenith angles and higher computational efficiency (He et al., 2024). To represent ice albedo, Whicker-Clarke et al. (2022) further developed SNICAR-ADv4 by integrating and extending key features from earlier radiative transfer models to achieve more accurate simulations of a spectrally resolved cryospheric column of snow and ice with a refractive boundary, while incorporating light-absorbing constituents (LACs), such as black carbon (BC) and algae, into this standalone radiative transfer model. It simulates bare ice using the physical microscopic structure of the ice, including the ice density, the scattering air bubbles within an absorbing ice medium, and a refractive boundary that depicts the refraction across snow-ice interfaces (Briegleb and Light, 2007; Gardner and Sharp, 2010; Mullen and Warren, 1988).

Nevertheless, the ice albedo is typically prescribed as constant values in the visible (VIS) and near-infrared (NIR) spectral regions in Earth system models. For instance, Ice albedo is 0.6 in the visible and 0.4 in the NIR in the default version of the Energy Exascale Earth System Model (E3SM) and the Community Earth System Model (CESM) version 2 (Whicker-Clarke et al., 2024). Prescribing constant albedo values does not represent the physical variability of solid ice albedo or its spectral changes under varying conditions. To advance ice radiative transfer modeling in Earth system models, Whicker-Clarke et al. (2024) incorporated SNICAR-ADv4 into the E3SM (specifically its land component, ELM), in which the GrIS ice physical properties are retrieved from the satellite observation data. This enhancement enables more realistic simulations of the GrIS bare ice albedo, and concurrently reveals that the default ELM method overestimates bare ice albedo by 4% in the visible and 7% in the NIR bands. However, the quality information of MODIS albedo products were not considered in the process of acquiring bare ice properties in their study. Schaaf et al. (2011) noted that the MODIS poor-quality inversions beyond a SZA of 70° are characterized by high noise and often significantly lower than the more stable and consistent values observed at smaller SZAs. Omitting quality flags could, therefore, lead to an underestimation of Greenland's snow/ice albedo and introduce significant uncertainties in the retrieval of bare ice physical properties. Despite the aforementioned modeling advances, the Common Land Model (CoLM) still uses fixed values to represent ice albedo (0.60 in the visible and 0.40 in

the NIR). For the purpose of investigating the impacts of bare ice metamorphism under polar
warming, it is also imperative to incorporate ice radiative transfer techniques into CoLM to
enhance albedo modeling with more realistic and physical representations of
snow-ice-LAC-radiation interactions.

In this study, we focus on the bare ice region of the GrIS, characterized by the presence of
land ice, and bare ice is exposed by snow melting during the ablation season. The aim of this
study is to develop a more reliable dataset of Greenland's bare ice physical properties by
incorporating the quality information of MODIS albedo products, and explore the bare
ice-albedo feedback associated with the metamorphism of bare ice after the implementation
of the SNICAR-ADv4 into the CoLM. This paper is organized as follows. Section 2 provides
descriptions of the CoLM snow and ice albedo schemes and details the model simulations, as
well as the explanation of the use of various MODIS products to inform the ice albedo
calculations in SNICAR-ADv4. Section 3 compares the differences in albedo simulations
with and without ice radiative transfer solver (SNICAR-AD and SNICAR-ADv4), and
quantifies the impact of varying bare ice properties on the near-surface air temperature and
the snow cover. Section 4 is conclusions and discussion.

**2. Models, Data, and Methods**
**2.1 Snow and Ice Albedo Schemes**
This study utilizes two distinct implementations of the SNICAR model within the CoLM for
snow and ice albedo simulations: (i) the baseline SNICAR-AD version (Dang et al., 2019)
and (ii) the enhanced SNICAR-ADv4 version (Whicker-Clarke et al., 2024). Both versions
adopt identical snow albedo algorithms but exhibit distinct ice albedo treatments. Specifically,
as shown in Figs. 1a and b, the SNICAR-ADv4 accounts for radiative transfer through the ice
column, while the SNICAR-AD prescribes ice albedo as constant values: 0.6 for visible (VIS:
0.3–0.7 μm) and 0.4 for near-infrared (NIR: 0.7–5.0 μm) bands. The snow albedo scheme of
SNICAR-AD/SNICAR-ADv4 in the CoLM computes snow albedo for the multi-layer (up to
5 layers) snowpack with the two stream radiative transfer scheme of the delta-Eddington
approximation and adding-doubling technique, accounting for the effects of snow properties
(e.g., grain size and shape) and LAC contamination on snow albedo.

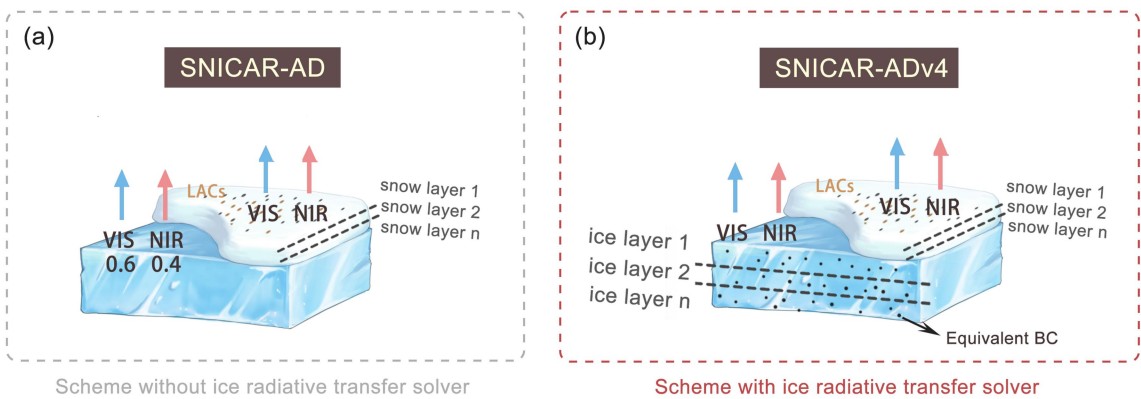


**Figure 1.** Schematic representation of the snow and land ice column in CoLM SNICAR-AD
and SNICAR-ADv4.

For snow albedo simulation, the SNICAR-AD/SNIACR-ADv4 embedded in CoLM uses the
physical properties of the snowpack and albedo of the top layer of the underlying ground to
determine the column albedo (Flanner and Zender, 2006). The input variables consist of
direct and diffuse radiation, the surface downward solar spectrum, the solar zenith angle (for
direct radiation), the ground albedo beneath the snowpack, vertical profiles of snow grain size,
snow layer thickness and density, aerosol concentrations of each snow layer, as well as the
optical properties of both snow and aerosols. Meanwhile, parameterizations for snow grain
shapes (sphere, spheroid, hexagonal plate, and Koch snowflake) and LACs-snow mixing
states (BC/dust externally or internally mixed with snow grains) are included to improve the
simulations of snow surface energy and water balances (Hao et al., 2023).

For ice albedo modeling, the advent of the SNICAR-ADv4 enables us to explore the regional
climatic response induced by changes in ice albedo using the ice microphysical properties
derived from satellite products. As proposed by Whicker-Clarke et al. (2024), the radiation
transfer process within the ice layer can be calculated in the land surface model, which
requires input variables such as ice density, air bubble effective radii within the ice,
equivalent BC concentrations, and downward solar spectra. The need for air bubble
parameters arises from the representation of ice layers as collections of independently
scattering air bubbles within a solid ice medium in SNICAR-ADv4, while snow layers are
treated as independently scattering ice crystals in an air medium (Picard et al., 2016;
Whicker-Clarke et al., 2022).

In addition to the albedo scheme, we briefly describe the physical processes represented by
the glacier and snow modules in CoLM to clarify model capabilities. The glacier component
is designed to capture essential surface thermodynamic processes, including full surface
energy balance calculations and subsurface heat diffusion through a multi-layer ice column.
However, it omits several key elements found in more advanced land ice schemes: (1) the
model assumes fixed ice thickness and does not track accumulation or ablation, lacking
mass-conserving SMB computation; (2) glacier geometry is static, with no coupling to an ice
sheet model for dynamic evolution; and (3) meltwater generated from glacier ice is retained
rather than routed to runoff, leading to unrealistic surface water storage. In contrast, the snow
component in CoLM simulates several critical processes: (1) multi-layer snowpack energy
and mass balance, including radiative, turbulent, and conductive heat fluxes; (2) vertical
snow temperature evolution and phase changes; (3) melt, liquid water percolation, refreezing,
sublimation and snowmelt runoff generation; and (4) snow aging and albedo evolution, with
consideration of the impacts of LAPs, as represented by SNICAR-AD/SNICAR-ADv4.

**2.2  Data**
MODIS MCD12C1, MOD09CMG, and MOD10C1 products with consistent 0.05° spatial
resolution were utilized for GrIS bare ice monitoring during the summer melt seasons of
2000-2020. The MCD12C1 Version 6.1 annual land cover type product (Friedl et al., 2010)
provided initial cryospheric classification by excluding grids not categorized as snow or ice.
The MOD09CMG (Vermote 2021) band 2 reflectance (0.841–0.876 μm) was employed for
bare ice-snow discrimination, where pixels with reflectance values below 0.6 were classified
as bare ice. Comparative spectral analysis of MODIS imagery by Shimada et al. (2016)

revealed markedly greater surface reflectance in snow-covered pixels relative to bare ice across all spectral bands, with maximal contrast observed at 0.86 μm. The robustness of this threshold was confirmed by Antwerpen et al. (2022) through comparison with Landsat 8 OLI (Operational Land Imager), showing a relative error of only 0.16% in bare-ice classification accuracy. The MOD10C1 product was further used to exclude pixels with cloud obstruction percentage exceeding 90% or snow cover fraction above 90% (Antwerpen et al., 2022; Whicker-Clarke et al., 2024). The derived bare ice extent was filtered by excluding pixels above the mean equilibrium line altitude of 1679 m a.s.l., defined as the 95th percentile of ablation zone elevations (Antwerpen et al., 2022). This conservative threshold minimizes sporadic high-elevation detections while maintaining robust estimation of the mean equilibrium line altitude (Antwerpen et al., 2022).

The MODIS MCD43C3 product (Schaaf et al., 2002) is used to retrieve bare ice physical properties by using standalone SNICAR-ADv4, and to evaluate CoLM-simulated albedo over the GrIS bare ice regions. This daily product provides spectral (MODIS bands 1 to 7) and broadband (VIS 0.3–0.7 μm, NIR 0.7–5.0 μm and shortwave 0.3–5.0 μm) black-sky albedo (BSA) and white-sky albedo (WSA) at local solar noon, derived from 16 days of Aqua-Terra merged surface albedo dataset based on the bidirectional reflectance distribution function (BRDF) algorithm (Schaaf and Wang, 2021). Compared with the GLASS-AVHRR and C3S-v2 albedo products, MCD43C3 demonstrates superior performance for monitoring snow albedo, exhibiting the lowest bias and RMSE over snow and consistent performance across diverse snow cover conditions (Urraca et al., 2022). In the GrIS, MCD43A3 was found to outperform the GLASS albedo product and even the reconstructed albedo based on the MOD10A1, for the sites located in the GrIS ablation zone (Ye et al., 2023). In this study, shortwave albedo under direct radiation is treated as equivalent to the BSA, in accordance with the widely accepted terminology used in the MCD43C3 product.

Considering the little difference between BSA and WSA for a typical summer day, using BSA is considered acceptable for analyzing the GrIS during the summer (Alexander et al., 2014; Stroeve et al., 2005). The extracted variables in this study from MODIS MCD43C3

include Band 2 BSA, broadband BSA (visible, near-infrared and shortwave), along with local
noon solar zenith angles (SZAs) and albedo quality index. The MCD43C3 albedo quality
index helps identify regions with cloud cover contamination, detrimental atmospheric
conditions, or insufficient observational data. Figure 2a shows the daily variation of the
regionally weighted average SZA over Greenland during May-September. The period with
SZA>70° occurs primarily in September. For the relationship between the SZAs of
MCD43C3 and their spatiotemporally corresponding albedo quality index (Fig. 2b), it can be
seen that the percentage of low-quality indices (4 and 5) rises drastically as the SZA increases
at higher SZA. To ensure reliable satellite-retrieved bare ice physical properties, we excluded
all albedo values identified with a low-quality index (4 or 5), regardless of the SZA. Figure
2b shows that the proportion of low-quality indices increases markedly when the SZA
exceeds 70°, indicating that such filtering primarily affects high-SZA retrievals.

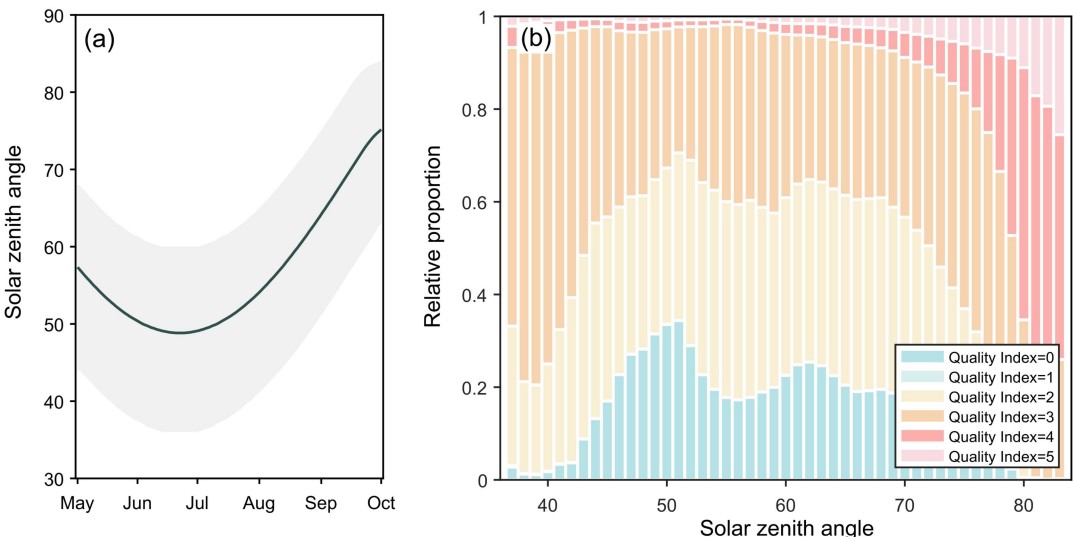


**Figure 2.** Regional-weighted mean SZAs of Greenland at local noon from May to September
(a; solid line). Grey shaded area represents the range of SZAs over Greenland. Relative
proportion of the quality index of MCD43C3 albedo dataset under different SZAs over
Greenland during May to September (b; 0 for best quality and 5 for poorest quality)

## 2.3  Parameter sensitivity of ice spectral albedo in SNICAR-ADv4

We use the standalone SNICAR-ADv4 model to briefly examine the key factors influencing
the spectral albedo of ice under direct illumination conditions. These factors include SZA, ice
density, air bubble effective radius (Reff), and black carbon (BC) concentration. This
sensitivity analysis provides a foundation for the subsequent method of obtaining ice physical
properties (Section 2.4). As shown in Fig. 3a, total internal reflection occurs at wavelengths
around 3μm for SZA greater than 55°, and the wavelength range for total internal reflection
expands with the increases in SZAs. This phenomenon occurs for pure and smooth ice
surfaces but is not representative of naturally occurring ice, which typically has impurities
and rough surfaces. For the dependency of albedo on ice density and air bubble effective
radius, the spectra show that the albedo declines as the ice density and air bubble radius
increases since air bubbles within the ice are responsible for the scattering light and smaller
bubbles scatter light more efficiently in the visible and near-infrared parts of the spectrum
(Figs. 3b-c). Furthermore, BC impacts ice albedo rather uniformly across the visible spectrum
and has almost no impact at $\lambda > 1.0$ μm (Fig. 3d).

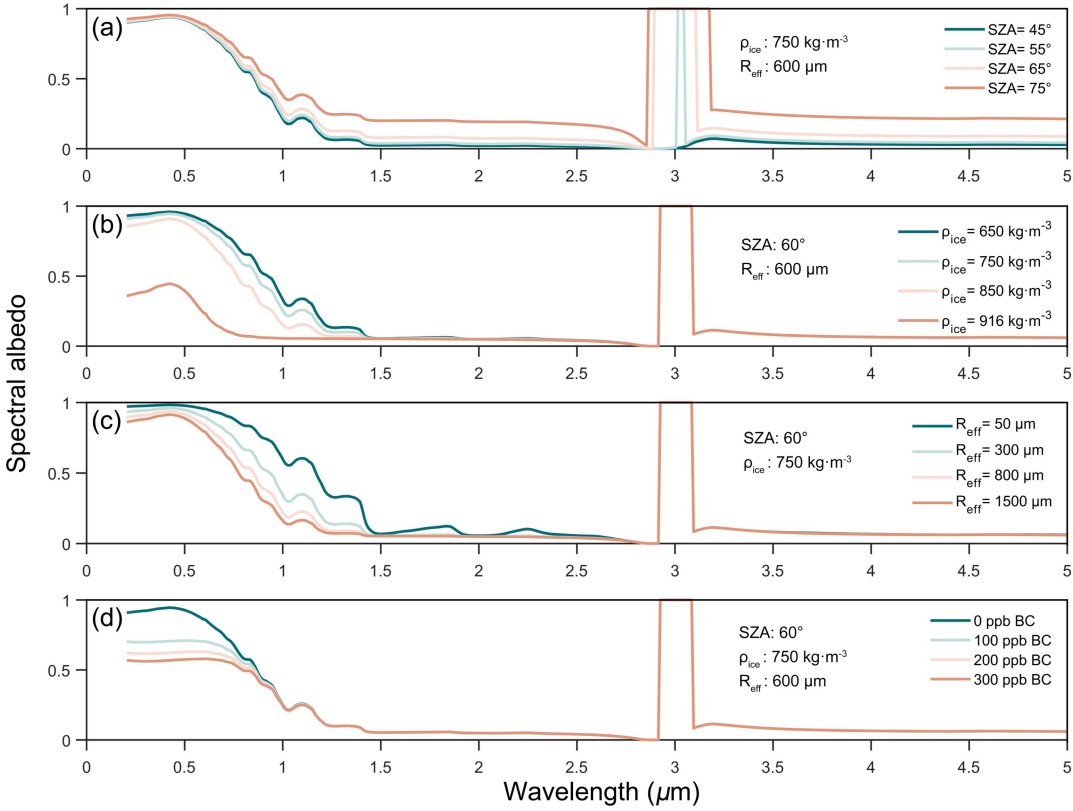


**Figure 3.** Spectral albedo simulated by standalone SNICAR-ADv4 under direct incident
irradiance with varing (a) SZA, (b) ice density, (c) air bubble effective radius and (d) BC
concentration.

While these controlled simulations clarify the fundamental optical behavior of ice under idealized conditions, natural environments involve more complex interactions shaped by microstructural evolution and meteorological forcing. A synthetic description of bare ice metamorphism includes the evolution of air bubbles and density: newly fallen snow starts with low density and open pore spaces, which become compacted through wind-driven grain fragmentation and rounding, forming wind slabs. Further densification occurs via grain-boundary sliding and pressure-induced deformation, during which air bubbles become sealed and gradually shrink under compression (Tedesco et al., 2016). In ablation zones, these densification processes are coupled with surface metamorphism. Glaciers undergoing melt often develop a porous weathering crust composed of loosely interlocked crystals, formed by differential solar absorption along grain boundaries, a process termed internal ablation (Müller and Keeler, 1969). Under overcast, windy, and warm conditions, this crust is preferentially removed, exposing denser, glazed ice beneath. Temperature-driven grain sintering and densification further reduce SSA by smoothing and coalescing ice grains (Flanner and Zender, 2006; Hofer et al., 2017). Concurrently, air bubble growth from differential solar heating and subsurface melting continues to modify the microstructure and optical properties of the ice.

## 2.4 Method

The method for obtaining ice physical properties (ice density, air bubble effective radius and equivalent BC) from MODIS bare ice albedo involves two main steps (Whicker-Clarke et al., 2024). First, as detailed in Section 2.2, bare ice spatiotemporal distribution was determined through the integrated use of MODIS products, employing MCD12C1 to exclude non-cryospheric pixels, MOD09CMG to distinguish bare ice from snow cover, and MOD10C1 to apply snow and cloud masking. Second, the bare ice physical properties (ice density and air bubble effective radius) are retrieved using the physical properties and SZA within the precomputed standalone SNICAR-Adv4 lookup table to match MCD43A3 band 2 BSA. Notably, this step derives only ice density and air bubble effective radius, whereas equivalent BC concentration requires additional processing steps described later in this

section. After obtaining all bare ice physical properties (ice density, air bubble effective

radius, and equivalent BC concentration), we upscaled the data from a spatial resolution of

$0.05°\times0.05°$ to $0.5°\times0.5°$.

The lookup table was generated using the standalone SNICAR-ADv4 radiative transfer

model by testing a range of parameter combinations within physically constrained ranges,

including ice density (650-916 kg·m⁻³) and air bubble radii (100-1500 μm), as well as the

SZAs spanning 35° to 75° to represent typical local noon conditions across the GrIS grid cells.

Following the SNICAR-ADv4 modeling configuration, ice with densities above 650 kg·m⁻³

is treated as bubbly ice, following the modeling approach in Whicker-Clarke et al. (2022),

which showed optimal agreement with in situ measurements. However, because the

density-bubble radius relationship for GrIS bare ice remains poorly constrained, we apply a

linear density-radius relationship as a first-order approximation for calculating the specific

surface area (SSA), where densities of 650 kg·m⁻³ and 916 kg·m⁻³ correspond to bubble radii

of 50 μm and 1500 μm, respectively (Fig. 4a). This parameterization is provisional and awaits

future observational validation. For each parameter combination, the band 2 albedo, SSA and

the volume fraction of air ($V_{air}$) were then output by the standalone SNICAR-ADv4. The SSA

is a measure of the total surface area of ice-air interfaces relative to the ice mass. The

relationship between the SSA (units: $m^2 \cdot kg^{-1}$) and ice density and air bubble effective radius

is given by Eq.1, where $\rho_{blk}$ is layer bulk ice density used to calculate the volume fraction of

air (Eq.2).

$$SSA = \frac{3V_{air}}{\rho_{blk}R_{eff}} \qquad (1)$$

$$V_{air} = \frac{\rho_{ice}-\rho_{blk}}{\rho_{ice}} \qquad (2)$$

Figure 4b shows the band 2 albedo from the SNICAR-ADv4 lookup table as a function of

SSA, illustrating that the modeled albedo is primarily determined by SSA rather than the

specific combination of ice density and bubble size. Consequently, the retrieval algorithm

selects the (density, radius) combination that most closely reproduces the observed Band 2

albedo. Since MCD43C3 provides the band 2 albedo and SZA for each bare ice grid cell, the

corresponding bare ice physical properties can be inferred from the lookup table. It is

important to note, however, that the resulting bare ice property maps (Figs. 4c-f) represent just one plausible solution among several combinations that could yield similar SSA and albedo values.

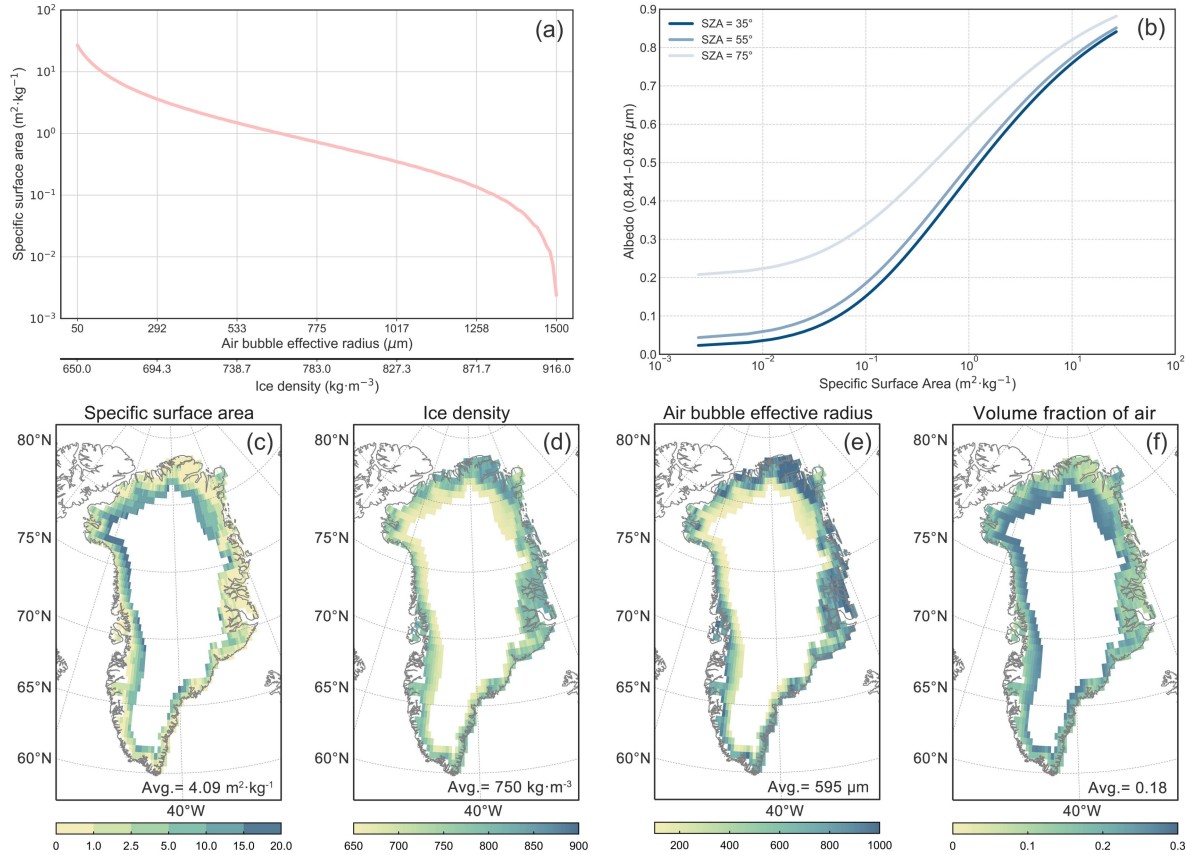

**Figure 4.** The relationship between ice specific surface area (SSA; $m^2 \cdot kg^{-1}$), air bubble effective radius (μm) and ice density (kg·m$^{-3}$) under a linear density-radius relationship (a first-order approximation) assumed in this study (a). MCD43C3 band 2 (0.841-0.876 μm) albedo as a function of SSA and solar zenith angle (b). Spatial distribution of JJA (c) specific surface area ($m^2 \cdot kg^{-1}$), (d) ice density (kg·m$^{-3}$), (e) air bubble effective radius (μm) and (f) volume fraction of air in the period of 2000-2020.

After acquisition of the daily ice density and air bubble effective radius of the GrIS (Figs. 4d and 4e), we again employed the standalone SNICAR-ADv4 model to simulate the NIR and visible albedo for each bare ice grid cell of the GrIS. Using an iterative optimization approach, we derived the equivalent BC concentration by adjusting the BC input parameter in the standalone SNICAR-ADv4 until its simulated visible albedo matched the MODIS MCD43C3

observations. This inversion method relies on the strong influence of LACs on visible albedo and their negligible impact on NIR albedo over bare ice (Schneider et al., 2019). As seen in Figs. 5a-c, there is minimal difference in the albedo in the NIR band, with a slight underestimation of 0.029 by the standalone SNICAR-ADv4. In contrast, the SNICAR-ADv4 significantly overestimated the visible albedo by up to 0.293 when using these bare ice properties, as it did not account for the LACs (Figs. 5d-f). We incrementally adjusted the input BC concentration in the standalone SNICAR model to match the visible albedo values from MCD43C3 data at each GrIS bare ice grid cell (Figs. 5h and i). This process yielded the daily equivalent BC concentrations shown in Fig. 5g. Based on the MODIS data and the standalone SNICAR-ADv4 lookup table, the daily 0.5-deg ice density, air bubble effective radius and equivalent BC data were then processed into monthly timescale as input for CoLM. Besides, it is worth mentioning that not all bare ice grid cells are informed by the bare ice physical properties data in each summer month. These grid cells are filled with the climatological mean values of bare ice physical properties when retrievals fail due to clouds or poor data quality.

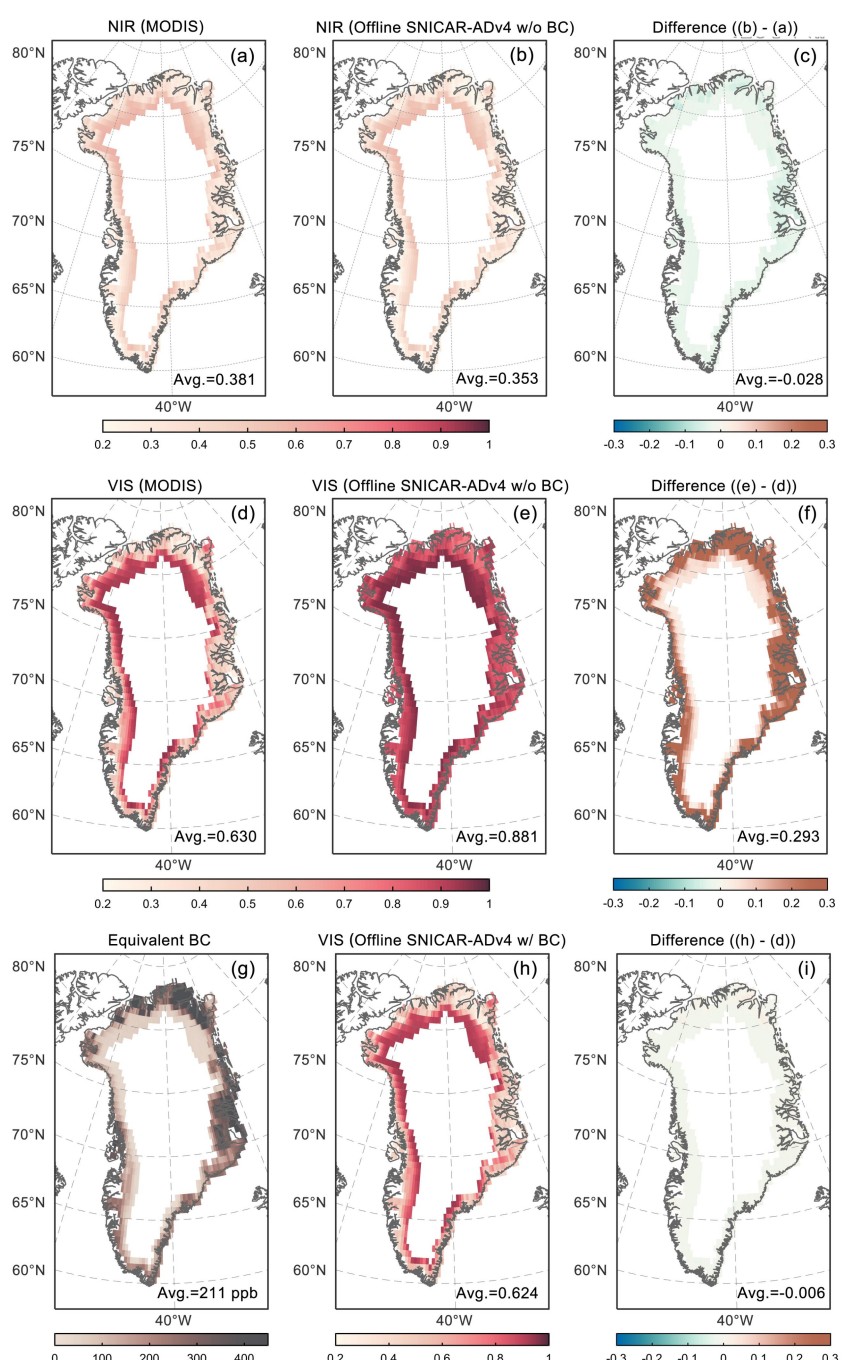

**Figure 5.** The spatial distributions of MODIS bare ice albedo and standalone SNICAR-ADv4 bare ice albedo excluding LACs in (a, b) near-infrared and (c, d) visible bands for JJA from 2000 to 2020, along with (c, f) their differences. The spatial distributions of (g) equivalent black carbon, (h) the standalone SNICAR-ADv4 bare ice visible albedo with equivalent black carbon (ppb), and (i) its difference from the MODIS bare ice visible albedo.

## 2.5 Model simulation

We conduct several offline CoLM simulations with the embedded SNICAR-ADv4 and SNICAR-AD schemes on a 0.25×0.25-degree resolution, driven by the atmospheric forcing from the hourly single-level surface dataset of the European Center for Medium-Range Weather Forecasts' fifth-generation atmospheric Reanalysis (ERA5) in the GrIS. Compared with other atmospheric forcings, ERA5's precipitation rates exhibit a higher correlation with measured net accumulation over the GrIS (Schneider et al., 2023). We run the model simulations for the years 1980–2020 and the summer melt season (June, July and August; JJA) during 2000-2020 is used for analysis. Aerosol concentration in the snow layer is calculated based on the prescribed monthly aerosol (BC, dust, OC) wet and dry deposition flux from the CESM2-WACCM simulations in CMIP6 experiments (Danabasoglu et al., 2020). The monthly bare ice properties for ice radiative transfer process are inferred from MODIS products using the standalone SNICAR-ADv4 over the bare ice region of the GrIS, covering JJA from 2000 to 2020, as the MODIS products has been available since 2000. To prevent possible unusual model behavior when shifting bare ice albedo schemes, the bare ice properties from the summer of 2000 were used in a brief spin-up run for the variable bare ice conditions in our experimental runs from 1998 to 2000. For land ice patches informed by the ice properties, the bare ice albedo is first calculated and replaces the constant values (0.6 for VIS and 0.4 for NIR). If snow is present over the ice, the new ice albedo of underlying ice column is used as the lower boundary to calculate snow albedo. The total patch albedo is then determined by the fractional coverage of land types and snow cover.

In this study, we analyzed output variables from three sets of CoLM simulations: (1) those using SNICAR-AD with fixed bare ice albedo (0.6 for visible and 0.4 for near-infrared), (2) those using SNICAR-ADv4 with annually-varying bare ice properties and (3) those using SNICAR-ADv4 with bare ice properties held constant at year 2000 values for all years. The simulations output two variable groups: (a) surface albedo (visible, near-infrared, and shortwave under direct radiation) and bare ice fraction for albedo evaluation; (b) 2-m temperature, snow cover fraction, and snow water equivalent to quantify the effect from the

bare ice metamorphism.
**3. Results**
**3.1 Mapping of GrIS bare ice physical properties**
Figures 4c-f display the spatial distribution of summer climatological mean of the bare ice
physical properties, including SSA, ice density, air bubble effective radius and volume
fraction of air. The bare ice density gradually decreases from the lower-elevation coastal
regions toward the interior, while the volume fraction of air shows an opposite pattern, as it is
calculated from the bulk ice-air mixture density and the density of pure ice (Figs. 4d and f;
Eqs. 1 and 2). SSA represents the total surface area of ice-air interfaces relative to the mass of
ice, determined by the volume fraction of air, effective diameter of air bubbles, and the bulk
density of the ice layer (Whicker-Clarke et al., 2022), with high values distributed in the area
along the mean equilibrium line (Fig. 4c). Given the large discrepancy in bare ice visible
albedo between the standalone SNICAR-ADv4 without LACs and the MCD43C3 in the
coastal regions of the GrIS (Figs. 5d and e), higher equivalent BC concentrations occur in
these areas compared to inland regions, indicating potentially more severe contamination,
particularly in the southeastern and northernmost parts of the GrIS (Fig. 5g)..

**3.2 Spatial and Temporal performance of CoLM Simulations**
In this study, the "land ice" area (Fig. 6a) refers specifically to glacier ice grid cells excluding
those persistently covered by snow. Therefore, interior regions of the GrIS, which are
continuously overlain by permanent snow cover, are not counted as "land ice" in this figure.
This explains the absence of land ice coverage in the GrIS interior. The land ice fraction in
Fig. 6a represents the proportion of glacier ice within a grid cell after filtering out areas
where snow cover fraction remains at 100%. Grid cells with land ice fraction < 1 contain a
mix of glacier ice and other surface types (e.g., bare soil). In contrast, Fig. 6b shows the
exposed bare ice fraction, which further considers seasonal snow cover using the
SNICAR-AD scheme. It is important to note that, despite a slight difference in snow cover
fraction simulations, the choice of snow albedo scheme does not affect the selection of bare
ice regions. The frequency distribution of the exposed bare ice fraction is shown in Fig. 6c.
The bare ice fraction ranges from 0 to 0.7 across the grid cells, with the majority of grid cells
exhibiting a bare ice fraction below 0.5. The histogram bars represent the relative proportion
of grid cells within each bare ice fraction interval.

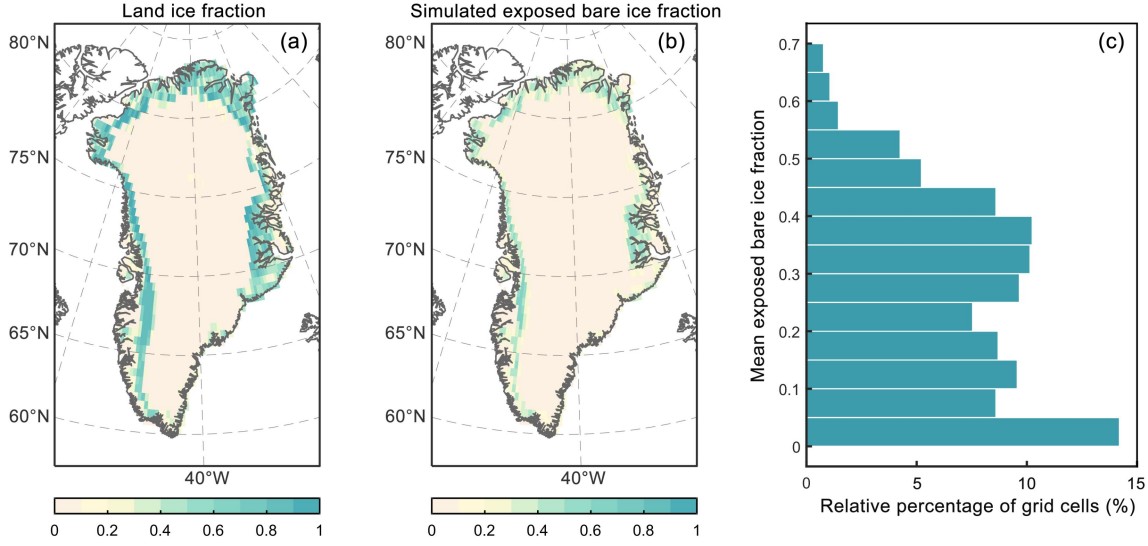


**Figure 6.** Spatial distribution of (a) the fraction of land ice underlying the snowpack,
excluding interior regions of the GrIS that remain fully snow-covered throughout JJA
(2000–2020), and (b) the mean exposed bare ice fraction during JJA over the same period,
based on snow cover simulated by CoLM using the SNICAR-ADv4 scheme. Panel (c) shows
the relative frequency distribution of mean exposed bare ice fraction, considering only grid
cells with nonzero bare ice exposure. Each bar indicates the percentage of these grid cells
whose mean bare ice fraction falls within a given interval, relative to the total number of bare
ice grid cells.

To assess whether the integration of an ice radiative transfer solver in CoLM improves albedo
simulations, we compared simulated albedo with the MCD43C3 albedo in shortwave, visible
and NIR regions of the spectrum during the summer of 2000-2020 in bare ice region (Fig. 7).
Both SNICAR-AD and SNICAR-ADv4 simulations use the same default snow albedo
configuration, which includes spherical snow grains, the adding-doubling radiative transfer
solver, and external mixing of BC/dust with snow. In other words, the differences in
simulated albedo between SNICAR-AD and SNICAR-ADv4 arise solely from their different
treatments of ice albedo, as the snow albedo configuration remains identical. As seen in Figs.
7a-c, it is obvious that the SNICAR-AD enabled CoLM albedo is significantly overestimated
across all bare ice regions, by 0.086 in shortwave, 0.078 in visible and 0.095 in NIR.
Compared with CoLM SNICAR-AD, the application of the SNICAR-ADv4 scheme reduced
the overestimation of albedo for all bands, by 38% in the shortwave, 50% in the visible and
28% in the NIR (Figs. 7d-f).

Furthermore, for each grid cell over the GrIS bare ice region, we computed the
root-mean-square error (RMSE) between the MODIS observed albedo and model-simulated
albedo (CoLM-SNICAR-AD/SNICAR-ADv4) time series (2000-2020, 21 summer values per
cell). Comparative analysis of the spatial distributions of correlation coefficients, RMSE, and
linear trends (Figs. S1-S3) reveals that CoLM-SNICAR-ADv4 outperforms
CoLM-SNICAR-AD across all evaluation metrics. These metrics were derived by comparing
the 21-year summer albedo time series from model simulations and MODIS observations at
each grid cell: correlation coefficients evaluate the phase similarity of interannual variations,
RMSE quantifies deviation magnitudes, and linear trends (obtained via least-squares
regression) capture interannual albedo changes. The comprehensive spatial evaluation
demonstrates consistent improvements in both the spatial pattern and quantitative
representation.

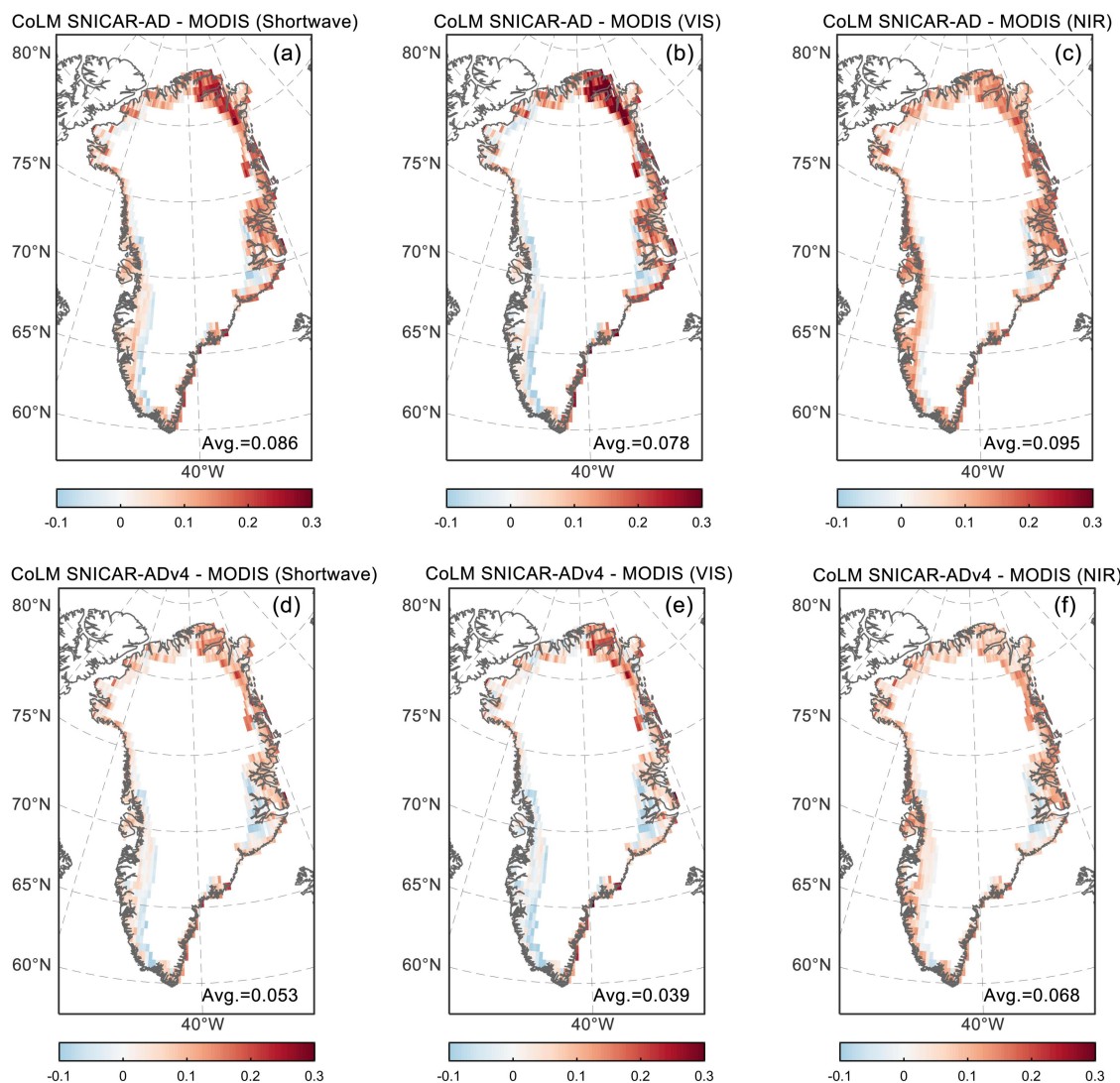

**Figure 7.** Spatial distribution of the difference of the 2000-2020 JJA albedo between the CoLM with different snow/ice albedo schemes (SNICAR-AD and SNICAR-ADv4) and the MCD43C3 in the (a, d) shortwave (0.3–5.0 μm), (b, e) visible (0.3–0.7 μm) and (c, f) near-infrared (0.7–5.0 μm) bands.

The decrease in the positive bias of CoLM SNICAR-ADv4 can also be clearly seen in the shortwave, visible and near-infrared albedo time series, with the area-weighted mean albedo of the GrIS bare ice regions steadily decreasing throughout the summer period from 2000 to 2020, compared with CoLM SNICAR-AD (Fig. 8). The albedo of CoLM SNICAR-ADv4 fluctuates around 0.47 in the shortwave, 0.53 in the visible, and 0.4 in the NIR, which is approximately 0.05 higher than the corresponding values in MCD43C3. In addition, the

484 CoLM SNICAR-ADv4 simulations exhibit synchronous variations in albedo with those of
485 MCD43C3, and there are relatively high temporal correlations between the CoLM
486 SNICAR-ADv4 and MCD43C4 albedo, with the values up to 0.95 for the shortwave, visible,
487 and NIR bands. In contrast, the albedo from the CoLM SNICAR-AD shows lower correlation
488 with MCD43C3 due to its constant ice albedo treatment. It is obvious that a large interannual
489 variability in the SNICAR-ADv4 enabled CoLM albedo is consistent with that of the
490 MCD43C3, while the simulated albedo using SNICAR-AD scheme presents a weaker
491 interannual variability. Regarding correlation with observations, SNICAR-AD achieves
492 slightly lower correlation (0.91) in the NIR band compared to its performance in the
493 shortwave and visible bands (both 0.92).

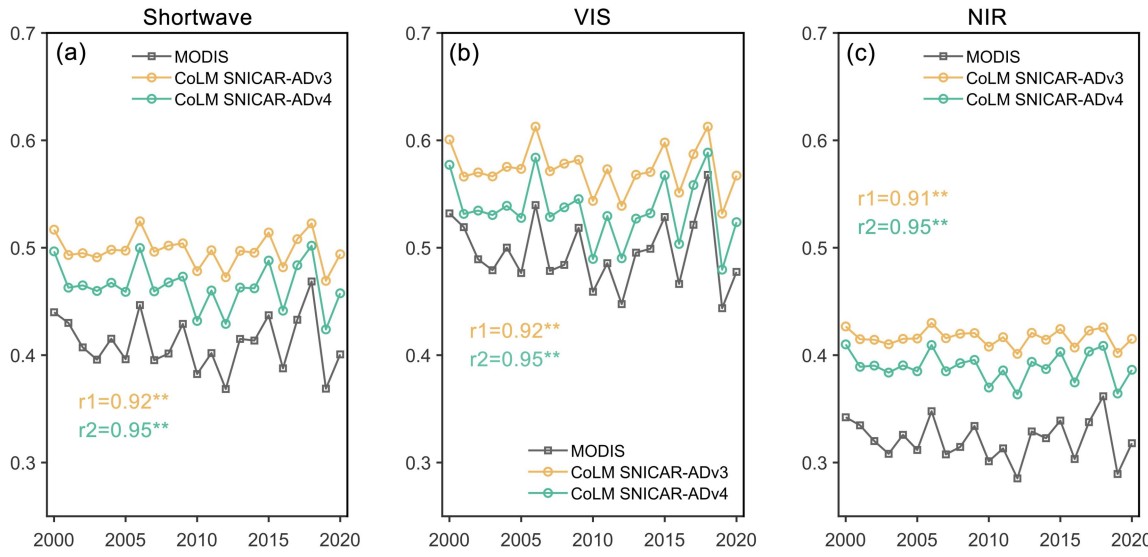

494

**Figure 8.** Time series of the 2000-2020 JJA CoLM SNICAR-AD and SNICAR-ADv4 albedo
versus the MCD43C3 albedo over bare ice region, in the (a) shortwave (0.3–5.0 μm), (b)
visible (0.3–0.7 μm) and (c) near-infrared (0.7–5.0 μm) bands. Double asterisks indicates
significance at the 99% confidence level.

499

500 Given that the bias reduction varies across regions with different bare ice coverages, we
501 explore the distribution of the albedo from CoLM SNICAR-AD, CoLM SNICAR-ADv4 and
502 MCD43C3 under different bare ice fractions. Generally, as bare ice fraction increases, CoLM
503 SNICAR-ADv4 can more effectively reduce the overestimation of shortwave broadband
504 albedo (BBA) compared to CoLM SNICAR-AD, due to its improved simulation of bare ice

BBA (Fig. 9). For regions where bare ice covers more than half the area, the albedo
overestimation of SNICAR-AD was reduced significantly by up to 99%. When the bare ice
fraction is between 0.4 and 0.5, the percentage of overestimation reduction in albedo
decreases to 74%, followed by regions with bare ice fraction of 0.3-0.4 (52%), 0.2-0.3 (38%),
0.1-0.2 (25%), and 0-0.1 (10%), respectively.

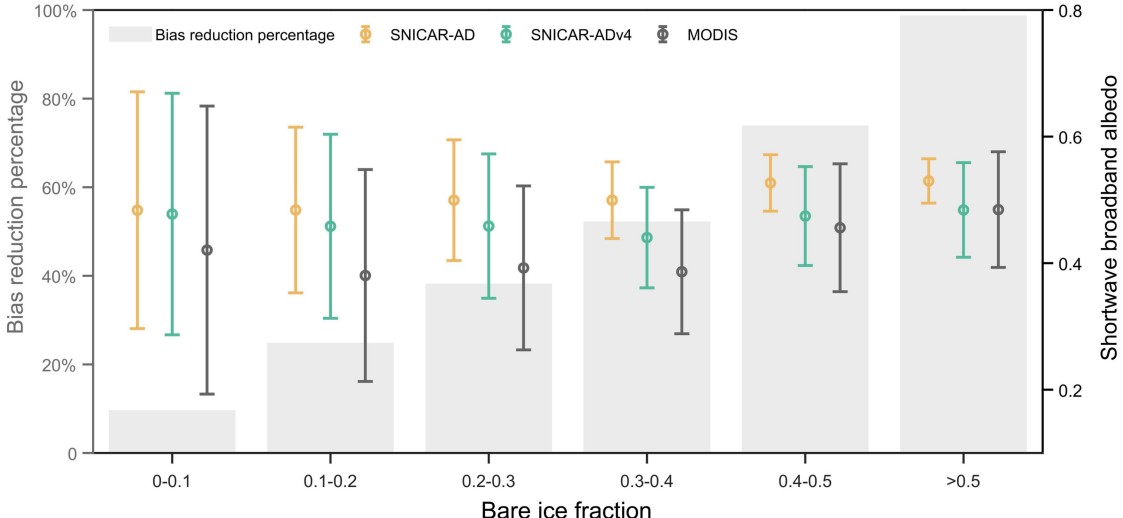


**Figure 9.** Mean shortwave broadband albedo from CoLM SNICAR-AD, CoLM
SNICAR-ADv4 and MCD43C3 under different bare ice fractions (error plots). The
uncertainty is calculated as double standard error, which reflects the 95% confidence interval.
The percentages of CoLM SNICAR-ADv4 albedo reduction in bias are represented by grey
bars.

## 3.3 A feedback revealed by bare ice property changes

The application of the SNICAR-ADv4 scheme in CoLM has significantly reduced the bias in
albedo simulations. To investigate the regional climatic response to bare ice metamorphism
of Greenland's bare ice region, we conduct a simulation in which the bare ice physical
properties for each year are set to the values from 2000. By calculating the difference in
simulated albedo between the simulations with annually varying bare ice properties and those
using the 2000 values, the model sensitivities to the change in summer bare ice albedo can be
assessed to quantify its impact on 2-m temperature and snow cover. To better highlight the
impact of changes in bare ice physical properties, the study area was restricted to regions with
a bare ice fraction larger than 0.4. Figures 10a-c compares the effects of bare ice
metamorphism on the 21-year summer mean albedo, 2-m temperature and snow cover
fraction, between simulations with annually varying bare ice properties and those using
constant year-2000 properties. The regional weighted mean albedo difference between the
two experiments reaches 0.032, indicating that the albedo in the bare ice region is reduced by
0.032 during the summer due to bare ice metamorphism (Fig. 10a). This leads to a 0.071°C
2-meter temperature forcing and a -0.011 change in snow cover fraction over the study period
(Figs. 10b and c). These results suggest that the temperature increase associated with the
change in albedo contributes to snow melting.

Spatially, the regions with strong response of near surface air temperature to bare ice albedo
changes are concentrated in the edge of the northwestern and western ablation zones, where
the 2-temperature increased by over 0.1°C in most part of these areas (Fig. 10b). A similar
response pattern can be also seen in the difference distribution of the snow cover (Fig. 10c),
with decrease in snow cover fraction exceeding 0.04 in parts of the northwestern and western
GrIS where temperature increases are most pronounced. To further evaluate the hydrological
implications of albedo-induced warming, we examined changes in snow water equivalent,
which integrates snow accumulation, meltwater retention, and sublimation processes. This
analysis indicates that bare ice metamorphism represented by annually varying ice properties
leads to a forcing that causes an average snow water equivalent decrease of 1.345 mm (Fig.
10d), consistent with the observed snow cover decline. The statistical distributions of changes
in 2-m temperature, snow cover, and snow water equivalent (Fig. 10e) reinforce the finding
that certain regions of the GrIS are especially sensitive to reductions in bare ice albedo.
Although the mean differences in 2-m air temperature, snow cover, and snow water
equivalent appear small, there are a considerable number of grid cells showing substantially
higher 2-m air temperature differences and notably lower snow cover and snow water
equivalent values. This indicates that certain regions of the GrIS exhibit relatively strong
sensitivity to changes in bare ice albedo (Fig. 10e). These coordinated changes manifest a
strong bare ice-albedo feedback in the GrIS bare ice region because bare ice albedo is
reduced through physical and biological melt-albedo processes that darken the ice surface as
the warming occurs in the ice surface.

The metamorphism of bare ice could be manifested in the changes in ice density and air
bubble radius with the ice, and these two factors jointly determine the specific surface area
(Eq.1) which have a one-to-one relationship with the bare ice albedo (Fig. 3b). From Fig. 10f,
the difference in BBA shows a strong positive correlation with the specific surface area, with
a correlation coefficient of 0.88 (significant at the 99% confidence level), since the two
simulations differ solely in their bare ice physical property inputs to the land surface model.
As more intense melt processes start in the early summer of the GrIS ablation zone after 2000,
the lower specific surface area, linked with the bare ice-albedo feedback, consistently
contributes to the reduction of the BBA (Fig. 10f). Additionally, according to the sensitivity
of modeled spectral albedo to the relevant parameters of the standalone SNICAR-ADv4
model (Fig. 3), the decreased bare ice albedo, associated with a lower specific surface area,
suggests an overall increase in ice density and a larger size of air bubbles within the ice in the
GrIS bare ice region.

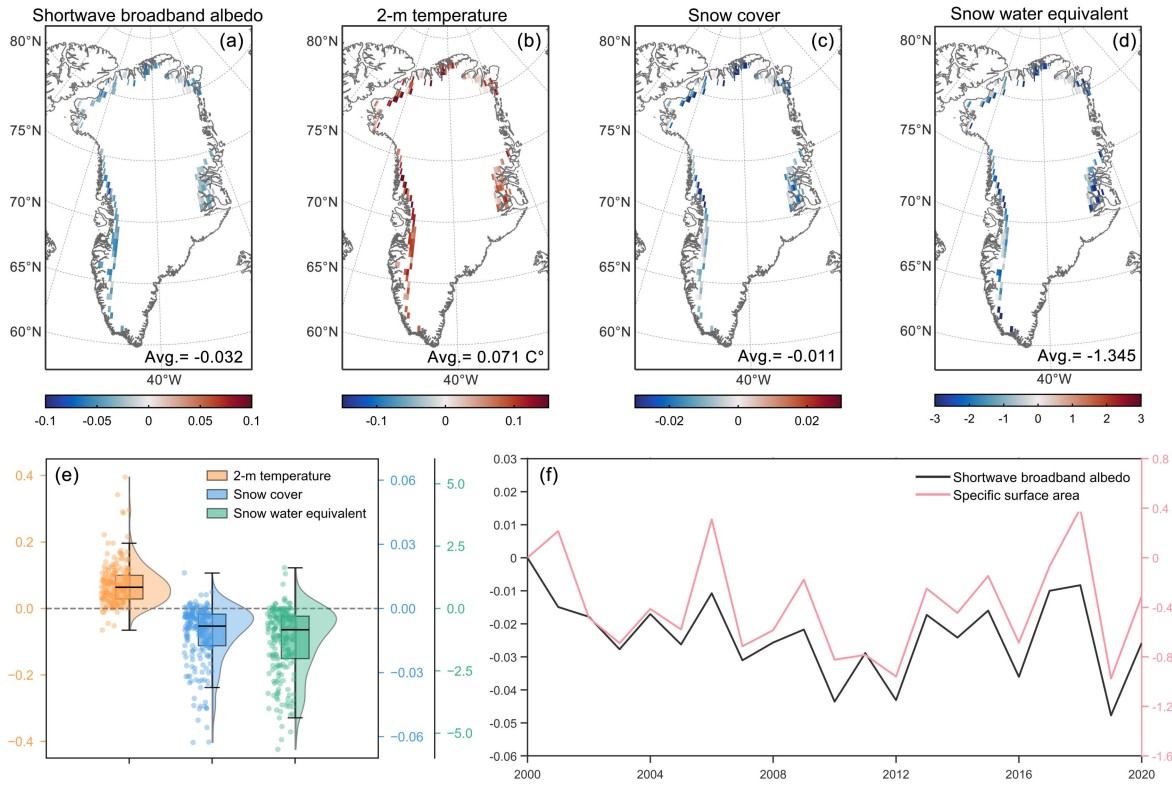

**Figure 10.** Spatial differences between simulations using annually varying bare ice properties and those using fixed year-2000 values during JJA (June–August) from 2000 to 2020: (a) surface albedo, (b) 2-m air temperature (°C), (c) snow cover fraction, and (d) snow water equivalent (mm). (e) Statistical distributions of differences in 2-m air temperature, snow cover, and snow water equivalent, shown using combined boxplots, left-side jittered points, and right-side half-violin plots. (f) Time series of differences in specific surface area (m²·kg⁻¹) and simulated shortwave broadband albedo between the two experiments.

After 2000, the metamorphism of bare ice in the Greenland bare ice region is mainly reflected in the decrease of SSA, which leads to ice darkening. This, in turn, induces regional near-surface temperature increases, causing snowmelt and ultimately resulting in a reduction of snow cover. Changes in snow cover directly determine the extent of bare ice exposure and significantly affect the albedo through snow-albedo feedback. The obvious snow cover reduction attributed to the changes in the physical properties of bare ice will cause more dark ice exposure and darkening, and make a constant contribution to albedo reduction in this ablation zone, suggesting a potential linkage between the bare ice-albedo and the

snow-albedo feedback.

**4. Conclusions and Discussion**
In this study, we incorporated SNICAR-ADv4 into the CoLM and made an enhanced
MODIS-informed bare ice physical properties to explore the response of the bare ice albedo
to ice metamorphism under polar warming. The application of SNICAR-ADv4, together with
the integration of MODIS-derived bare ice properties, significantly improved albedo
simulations by reducing the bias introduced by the default constant ice albedo treatment.
Specifically, bias reductions of 38%, 50%, and 28% were achieved for broadband, visible,
and near-infrared albedo, respectively. This improvement stems not only from the physically
enhanced radiative transfer calculations over the ice column in SNICAR-ADv4, but also from
the critical incorporation of MODIS-constrained ice optical properties, such as ice density
and bubble radius. These additions provide better physical realism and representation of
surface conditions across the bare ice zone. The snow and ice treatment used in CoLM
SNICAR-ADv4 and SNICAR-AD schemes are summarized in the Fig. 1, and it is evident
that SNICAR-ADv4 performs radiative transfer calculations not only over the snow column
but also over the ice column. During the summertime of 2000-2020, the bare ice BBA
decreased by 0.032 due to the changes in bare ice physical properties. The subsequent
darkening of the bare ice led to a 2-m air temperature forcing of 0.071°C, a change in snow
cover of -0.011 and snow water equivalent of -1.345 mm over the 21-year period, suggesting
that even a slight reduction in bare ice albedo can produce measurable climate responses in
the ablation region.

Our results are consistent with, and extend, recent progress in modeling bare ice albedo
modeling over the GrIS. Antwerpen et al. (2022) demonstrated that the regional MAR model
overestimated bare ice albedo by 22.8% below 70°N, leading to significant underestimation
of meltwater production. Similarly, Wicker-Clarke et al. (2024) found that the global
ELM-E3SM model overestimated shortwave broadband albedo by ~5% due to the use of
fixed albedo parameters, and showed that incorporating more realistic bare ice albedo
reduced the SMB by approximately 145 Gt between 2000 and 2021. Although both studies
focus on the GrIS, they differ in model structure and spatial resolution: MAR is a
high-resolution regional climate model, while ELM-E3SM is part of a coarser-resolution
global Earth system model. Despite these differences, both studies highlight a persistent
bias-systematic overestimation of bare ice albedo. The convergence of evidence from diverse
modeling frameworks underscores the need to improve bare ice representation in land surface
models. Building on these insights, our study examines the role of bare ice metamorphism,
particularly changes in specific surface area, in driving progressive surface darkening. By
isolating the feedback between evolving ice properties and surface energy balance, we
propose a physical mechanism for the observed albedo decline. Our sensitivity analysis
underscores how bare ice metamorphism can influence surface energy balance and the
importance of incorporating such processes in future model developments.

Our findings also highlight the role of the bare ice-albedo feedback linked to changes in ice
surface properties, as shown in Fig. 11. A marked reduction in snow cover occurred due to
lowered albedo in the ablation zone, exposing more bare ice and further reducing regional
albedo, especially in northern GrIS. This agrees with previous findings that increased bare ice
exposure has intensified the snow-albedo feedback in this region, with its strength rising by
51% from 2001 to 2017 (Ryan et al., 2019). The physical processes governing snowpack
evolution play a crucial role in modulating surface albedo and associated feedbacks,
particularly in the ablation zone of the GrIS, where snow loss accelerates bare ice exposure
and amplifies radiative forcing. More specifically, new snow quickly loses reflectivity
through grain growth and vapor diffusion, with subsequent changes driven by temperature
gradients and compaction (Flanner and Zender, 2006). Meltwater accelerates these processes
through melt-refreeze cycles (Brun 1989), creating a self-reinforcing system where both ice
exposure and snow aging enhance surface darkening. While biological and hydrological
factors such as algal growth play a secondary role in ice darkening (Ryan et al., 2019), our
results demonstrate that changes in bare ice properties, particularly a downward trend in
specific surface area at a rate of -0.007 $yr^{-1}$, exert a significant control over meltwater
production. We collectively term these processes of the variation in the bare ice albedo

associated with snow melting the bare ice-snow-albedo feedback (Fig. 11). As rising temperatures may further reduce ice albedo, this feedback could substantially increase Greenland's contribution to sea level rise through enhanced melting (Ryan et al., 2019), highlighting the need for improved process understanding in climate projections.

This study advances our understanding of the performances of the GrIS's snow and ice albedo simulations using different snow/ice schemes (SNICAR-AD and SNCIAR-ADv4), and the amplifying effect of bare ice on the albedo reduction through bare ice-snow-feedback mechanism. However, three key limitations constrain our current findings. First, the $0.5°×0.5°$ resolution is insufficient to accurately represent the narrow ablation zone, and big resolution gap between MODIS data and the model output is a limitation of this study. Second, CoLM's representation of GrIS glaciers prescribed fixed ice thickness and mass with internally retained meltwater prevents calculation of SMB, and excludes ice melt contributions to runoff. Although computationally efficient, this simplification systematically underestimates meltwater export from Greenland's ablation zones, where surface processes and especially meltwater runoff are the dominant contributors to mass loss (Ryan et al., 2019, van den Broeke et al., 2016, The IMBIE Team, 2020). Third, methodological constraints prevent independent quantification of ice density and air bubble size effects, as their relationship is prescribed in the lookup table ($\rho_{ice}$=650 kg·m$^{-3}$ corresponds to R$_{eff}$=100 μm, $\rho_{ice}$=916 kg·m$^{-3}$ corresponds to R$_{eff}$=1500 μm) based on the standalone SNICAR-ADv4 model. Future work will address these limitations by employing higher-resolution modeling for more precise delineation of bare ice margins, improving the representation of surface mass balance and ice-melt runoff through a more complete physically based snow and ice surface scheme, and establishing observational constraints on ice density and air bubble effective radius evolution to enhance ice albedo modeling. Future efforts are also needed to consider the actual LACs concentrations within the ice, including BC, dust and snow algae, rather than relying on equivalent BC, and evaluate their impacts on GrIS mass loss using fully coupled land-atmosphere models, which may reveal more pronounced feedbacks than offline simulations.

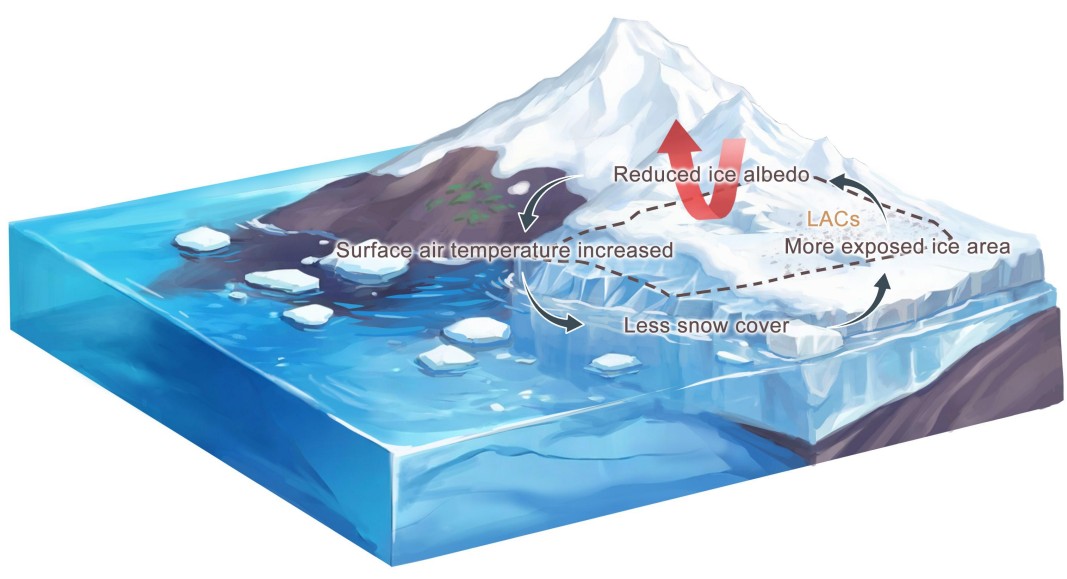

678

**Figure 11.** Illustration of the bare ice-snow-albedo feedback over the Greenland Ice Sheet. A reduction in ice albedo, primarily driven by changes in bare ice surface properties, exposes more bare ice, further lowering regional albedo and raising surface air temperatures. This leads to a decline in snow cover, which accelerates bare ice exposure and reinforces radiative forcing. This positive feedback loop intensifies melt, particularly in the ablation zone, contributing to increased surface darkening and meltwater production.

685

*Data availability*. The SNICAR-ADv4 enabled CoLM2024 code is available on GitHub at https://github.com/guoshuyang23/CoLM-SNICARADv4. The standalone SNICAR-ADv4 used in this study can be downloaded at https://github.com/chloewhicker/SNICAR-ADv4. MODIS snow cover data (MOD10C1) used to make the cloud and snow mask for bare ice pixels is available at https://nsidc.org/data/mod10c1/versions/6. MODIS surface reflectance data (MOD09CMG) used to retrieve the bare ice properties is from https://doi.org/10.5067/MODIS/MOD09CMG.061. MODIS surface albedo data (MCD43C3) used to evaluate the simulations and retrieve the bare ice properties is from https://doi.org/10.5067/MODIS/MCD43C3.061


*Author contributions*. SYG designed the study and wrote the paper. YJD was responsible for to conceptualization, supervision, and funding acquisition. HY contributed to revisions of the

manuscript. HBL provided technical support.

*Competing interests*. The contact author has declared that neither they nor their co-authors
have any competing interests.

*Acknowledgements*. We thank Chloe A. Whicker-Clarke for sharing the method for
processing ice optical property files in the standalone SNICAR-ADv4 for use in land surface
models. We acknowledge the two anonymous reviewers for their constructive comments that
substantially improved our paper.

*Financial support*. This research was funded by the Guangdong Major Project of Basic and
Applied Basic Research (2021B0301030007), the Natural Science Foundation of China
(under Grants U2342227, 42075160, and 42088101), the Southern Marine Science and
Engineering Guangdong Laboratory (Zhuhai) (No. SML2023SP216), and the specific
research fund of the Innovation Platform for Academicians of Hainan Province
(YSPTZX202143).

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
