# Peer review of "Enhanced MODIS-derived ice physical properties within CoLM revealing bare ice-snow-albedo feedback over Greenland Shuyang Guo1, Yongjiu Dai1\*, Hua Yuan1, Hongbin Liang1 Southern Marine Science and Engineering Guangdong Laboratory (Zhuhai), School of Atmospheric Science, Sun Yat-sen University, Zhuhai, China"

_EGUsphere, 2025_

## Author Comment (AC1)

**Responses to reviewer's comments about the manuscript, entitled "Enhanced MOIDS-derived ice physical properties within CoLM revealing bare ice-snow-albedo feedback over Greenland" (EGUSPHERE -2025-230)**

**General comments**

This paper examines the extent to which accounting for the physical properties of ice in areas of exposed bare ice on the Greenland ice sheet affects the albedo and surface air temperature as well as the extent of snow cover via what they call the ice-snow-albedo feedback. The work is based on the use of a SNICAR-ADv4 radiative transfer model (which explicitly represents the optical properties of snow and ice, taking into account several species of light-absorbing constituents) implemented in the CoLM surface model. It also takes advantage of MODIS products combined with data quality-indices to provide more reliable physical properties of bare ice that are used as inputs to SNICAR-ADv4. The simulation results are compared with those from an earlier version of SNICAR (SNICAR-AD), which uses constant ice albedo values. Comparison of the results from the two SNICAR versions allows to assess the importance of changes in ice properties (i.e. bare ice metamorphism) on the albedo and the surface climate.

The method does not appear to be novel as it is similar to that proposed by Wicker-Clarke et al. (2024), albeit with the Energy Exascale Earth System Model (E3ESM) rather than CoLM. A similar study has also been conducted by Antwerpen et al. (2022). You mention that you added quality-information regarding MODIS products. However, both Wicker-Clarke et al. (2024) and Antwerpen et al. (2022) excluded some pixels from the analysis and filtered data. Wouldn't this be a way of adding quality information? However, I acknowledge that the evolution of Greenland is a growing matter of concern with increasing mass losses now dominated by changes in surface mass balance (SMB). SMB is strongly dependent on surface albedo which is expected to decrease in response to surface meting and increase in the extent of darken areas. It is therefore of primary importance to investigate the response of a variety of models to surface processes including bare ice metamorphism. This is why, I recommend the publication of this paper after major (and minor) comments (see below) have been addressed.

**Responses:** We greatly appreciate your thorough evaluation and constructive feedback on our study. They are very helpful for improving our manuscript. We carefully revised the manuscript according to these comments. Our point-by-point responses are detailed below.

**Major comments**

1/ First of all, I found that the methodology is not sufficiently explained. This is detrimental for the overall understanding of the paper. I had to read the paper several times. I had to read Section 2 several times to understand the whole procedure. In my opinion, part of the problem is that the description of the method closely resembles that described in Wicker-Clarke et al (2024) but with the removal of a certain amount of information that would have been necessary for a full understanding of the method. More details should also be given about the different models used in this study. A number of things are not very clear:

i/ Which variables are simulated by CoLM and and for this study? It seems to me that this is not clearly stated anywhere. Is it albedo, but I thought that the albedo was calculated by SNICAR?

**Responses:** Thanks for your questions. In this study, we analyzed output variables from three sets of CoLM simulations: (1) those using SNICAR-AD with fixed bare ice albedo (0.6 for visible and 0.4 for near-infrared), (2) those using SNICAR-ADv4 with annually-varying bare ice properties and (3) those using SNICAR-ADv4 with fixed bare ice properties (2000 values). The simulations output two variable groups: (a) surface albedo (visible, near-infrared, and shortwave under direct radiation) and bare ice fraction for albedo evaluation; (b) 2-m temperature, snow cover fraction and snow water equivalent to quantify the effect from the bare ice metamorphism. These clarifications have been incorporated into Section 2.4 (Model Simulations) of the revised manuscript.

All of the aforementioned variables are output by the CoLM (Figs. 5-9 of the original manuscript), while the variables such as band 2, visible and near-infrared albedo from standalone SNICAR-AD (Figs. 2 and 3 of the original manuscript) are solely used for generating lookup tables and obtaining MODIS-informed bare ice physical properties of the GrIS.

ii/ Why is the BATS scheme mentioned (at the same level as SNICAR) whereas you never refer to in the rest of the paper? Mentioning the BATS scheme adds to the confusion.

**Responses:** Thanks for your question. The reference to the BATS scheme was inadvertently included in our initial draft but does not contribute to the current analysis of our study. We have now removed all mentions of BATS throughout the

manuscript to eliminate any potential confusion and maintain focus on the CoLM SNICAR-AD/SNICAR-ADv4.

My recommendation is therefore to clearly explain the functionalities of each of the models used in this study: SNICAR-AD, SNICAR-ADv4 and CoLM. In fact, in the current version of the paper, I have the feeling that the information is diluted in various places or that it arrives too late. To make things clearer, a scheme similar to that of Figure 10 could be incorporated in Section 2 (obviously without the panel illustrating the ice-snow-albedo feedback).

**Responses:** We sincerely appreciate this valuable suggestion. To clarify, we have thoroughly restructured Section 2 to provide concise, focused descriptions of SNICAR-AD, SNICAR-ADv4, and CoLM functionalities. Detailed revisions are presented in Section 2 (beginning on page 5 of this file). We relocated the model schematic (originally Fig. 10) to Section 2 as the new Fig. 1, now excluding the ice-snow-albedo feedback components. These modifications allow readers to immediately understand how CoLM SNICAR-AD and SNICAR-ADv4 simulate snow and ice albedo without consulting later sections.

This would also offer the opportunity to briefly present the physical processes associated with the evolution of the snowpack, such as compaction and refreezing, among others (see for example Flanner and Zender, 2005). This aspect is important because it is involved in the ice-snow-albedo feedback highlighted in the present paper.

**Responses:** Thank you for the valuable suggestion. The physical processes associated with the evolution of the snowpack, such as compaction and refreezing, are indeed important components of the ice-snow-albedo feedback. These processes were covered in the second-to-last paragraph of Section 4 Conclusions and Discussion (the font color was marked in red), as shown below:

Our findings also highlight the role of the bare ice-albedo feedback linked to changes in ice surface properties, as shown in Fig. 11. A marked reduction in snow cover occurred due to lowered albedo in the ablation zone, exposing more bare ice and further reducing regional albedo, especially in northern GrIS. This agrees with previous findings that increased bare ice exposure has intensified the snow-albedo feedback in this region, with its strength rising by 51% from 2001 to 2017 (Ryan et al., 2019). The physical processes governing snowpack evolution play a crucial role in modulating surface albedo and associated feedbacks, particularly in the ablation zone

of the GrIS, where snow loss accelerates bare ice exposure and amplifies radiative forcing. More specifically, new snow quickly loses reflectivity through grain growth and vapor diffusion, with subsequent changes driven by temperature gradients and compaction (Flanner and Zender, 2006). Meltwater accelerates these processes through melt-refreeze cycles (Brun 1989), creating a self-reinforcing system where both ice exposure and snow aging enhance surface darkening. While biological and hydrological factors such as algal growth play a secondary role in ice darkening (Ryan et al., 2019), our results demonstrate that changes in bare ice properties, particularly a downward trend in specific surface area at a rate of -0.007 yr$^{-1}$, exert significant control over meltwater production. We collectively term these processes of the variation in the bare ice albedo associated with snow melting the bare ice-snow-albedo feedback (Fig. 11). As rising temperatures may further reduce ice albedo, this feedback could substantially increase Greenland's contribution to sea level rise through enhanced melting (Ryan et al., 2019), highlighting the need for improved process understanding in climate projections.

**References:**

Flanner, M. G., and Zender, C. S. Linking snowpack microphysics and albedo evolution, J. Geophys. Res., 111(D12), https://doi.org/10.1029/2005JD006834, 2006.

Brun, E.: Investigation of wet-snow metamorphism in respect of liquid-water content, Ann. Glaciol., 13, 22–26, https://doi.org/10.3189/S0260305500007635, 1989.

Ryan, J. C., Smith, L. C., van As, D., Cooley, S. W., Cooper, M. G., Pitcher, L. H., and Hubbard, A.: Greenland Ice Sheet surface melt amplified by snowline migration and bare ice exposure, Sci. Adv., 5(3), eaav3738, https://doi.org/10.1126/sciadv.aav3738, 2019.

In the introduction, you also mention that the vertical profile of snow grain size as well as snow thickness are considered as input variables of the SNICAR model. If so, where do these input data come from? On the other hand, it seems to me that the snowpack model should be able to simulate these variables itself. They should therefore be considered as output variables. Can you clarify or comment please?

**Responses:** We sincerely appreciate your insightful comments. You are absolutely correct that these variables are simulated within CoLM's snowpack physics routines. To clarify their dual roles: (i) they are prognostic variables calculated by the snowpack model at each timestep, while (ii) functionally serving as inputs to the embedded SNICAR radiative transfer subroutine for snow albedo computations. This

integrated architecture ensures physical consistency between snow evolution and albedo calculations.

Overall, I suggest to reorganize Section 2 (while addressing the above comments) as follows: Section 2.1: Snow and ice albedo schemes / Section 2.2: Data / Section 2.3: Method / Section 2.4: CoLM simulation

**Responses:** We sincerely appreciate your insightful suggestions for improving the manuscript's organization. Following your recommendation, we have completely restructured Section 2 as follows:

[revised manuscript text omitted]

L130 (and also L154 and L203): You mention that ice albedo is 0.80 and 0.55 for VIS and NIR spectra. These values correspond more to the albedo values for fresh snow than to the albedo values for bare ice. Do SNICAR-AD-CoLM simulations actually use these values for bare ice? If so, it is not surprising that the use of SNICAR-ADv4 leads to a significant reduction in albedo. If this is the case, you should redo a SNICAR-AD-CoLM simulation with values more characteristic of those for bare ice.

Furthermore, Figure 10 shows that the ice albedo values for SNICAR-AD are 0.6 and 0.4 for VIS and NIR respectively (which seems more realistic to me). Please clarify

**Responses:** Thank you for your insightful comments regarding the bare ice albedo parameters. In response, we have updated all CoLM SNICAR-AD simulations with more realistic bare ice albedo values (0.60 for VIS, 0.40 for NIR), which now properly align with the original Figure 10. The revised results (Figs. R6 and R7) demonstrate that CoLM SNICAR-ADv4 still reduces the albedo overestimation compared to the CoLM SNICAR-AD, even after this parameter adjustment. All relevant sections of the manuscript and supplementary materials have been updated accordingly.

[Figure]

Figure R6. Spatial distribution of the difference of the 2000-2020 JJA albedo between the CoLM with different snow/ice albedo schemes (SNICAR-AD and SNICAR-ADv4) and the MCD43C3 in the (a, d) shortwave (0.3–5.0 μm), (b, e) visible (0.3–0.7 μm) and (c, f) near-infrared (0.7–5.0 μm) bands.

[Figure]

Figure R7. Time series of the 2000-2020 JJA CoLM SNICAR-AD and SNICAR-ADv4 albedo versus the MCD43C3 albedo over bare ice region, in the (a) shortwave (0.3–5.0 μm), (b) visible (0.3–0.7 μm) and (c) near-infrared (0.7–5.0 μm) bands. Double asterisks indicates significance at the 99% confidence level.

2/ My second comment is related to the effect of bare ice metamorphism on the surface ai temperature (+ 0.071°C) and on the reduction of ~1% of the snow cover. This does not seem very significant. To be more convincing, I recommend to provide additional diagnostics. As SNICAR (AD and Adv4) includes a snow scheme, I guess that all the elements are available for computing the surface mass balance and the runoff coming. This should help better quantify the actual impact of a more realistic calculation of the ice albedo.

**Responses:** Thank you for this insightful comment. We fully agree that further diagnostics would help clarify the physical implications of albedo changes and appreciate your suggestion regarding surface mass balance (SMB) and runoff. In response, we first revised the manuscript wording to more accurately reflect the spatial variability of the 2-m air temperature and snow cover changes, replacing terms like "pronounced" with "significant" where appropriate.

In addition, given the limitations of the current model, SMB and glacier runoff cannot be explicitly computed. The CoLM framework incorporates a thermodynamic glacier scheme that assumes fixed ice thickness and mass. Within this scheme, meltwater generated (if it occurs) from glacial ice is assumed to be fully retained in the ice column and does not contribute to runoff. This assumption precludes the SMB calculation and affects runoff interpretation. To better quantify the hydrological

implications within these constraints, we conducted a diagnostic analysis of snow water equivalent (SWE), which integrates processes such as snow accumulation, retention of meltwater, and sublimation. The results indicate an average SWE reduction of 1.345 mm due to the bare ice metamorphism represented by annually varying ice properties, consistent with the decrease in snow cover. To further highlight the regional variability and covariation of key variables, we introduce a new panel (Fig. R8e) that presents the statistical distributions of differences in 2-m air temperature, snow cover, and SWE using combined boxplots, jittered points, and half-violin plots. These distributions reveal that certain regions are strongly affected by the reduced albedo. Together, these coordinated changes highlight a pronounced bare ice-snow-albedo feedback, in which darkening of the bare ice surface leads to amplified warming and accelerated snow cover depletion.

[Figure]

Figure R8. Spatial differences between simulations using annually varying bare ice properties and those using fixed year-2000 values during JJA (June–August) from 2000 to 2020: (a) surface albedo, (b) 2-m air temperature (°C), (c) snow cover fraction, and (d) snow water equivalent. (e) Statistical distributions of differences in 2-m air temperature, snow cover, and snow water equivalent, shown using combined boxplots, left-side jittered points, and right-side half-violin plots. (f) Time series of differences in specific surface area (m²·kg⁻¹) and simulated shortwave broadband albedo between the two experiments.

Although the runoff analysis is precluded by model limitations, we examined the diagnostic output of surface runoff and found a counterintuitive reduction in runoff in most regions under the low albedo scenario. This is driven by: (i) retention of glacial meltwater in the ice layer, (ii) earlier snow depletion shortening the runoff season, and (iii) enhanced sublimation dominating local mass loss. Due to the structural limitations of the CoLM glacier scheme, we chose not to pursue detailed runoff analysis further.

We have revised Section 3.3 to incorporate these findings, and updated Section 4 to clarify the limitations associated with the glacier hydrology scheme. These updates emphasize the need for future work to couple the CoLM with a dynamic ice sheet model, which would allow more accurate SMB and ice-melt runoff simulations and thus provide a more complete picture of the GrIS mass loss response to albedo changes.

3/ The Discussion section lacks a detailed comparison with the results of Antwerpen et al. (2022) and Wicker-Clarke etal. (2024).

**Responses:** Thank you for your suggestion to strengthen the discussion by integrating a detailed comparison with Antwerpen et al. (2022) and Wicker-Clarke et al. (2024). In response, we have expanded our Section 4 ("Conclusions and Discussion") to include a detailed analysis of how our results relate to the findings of Antwerpen et al. (2022) and Wicker-Clarke et al. (2024). The additional content, now forming a new paragraph, has been inserted following the first paragraph of Section 4, as shown below:

Our results are consistent with, and extend, recent progress in modeling bare ice albedo modeling over the GrIS. Antwerpen et al. (2022) demonstrated that the regional MAR model overestimated bare ice albedo by 22.8% below 70°N, leading to significant underestimation of meltwater production. Similarly, Wicker-Clarke et al. (2024) found that the global ELM-E3SM model overestimated shortwave broadband albedo by ~5% due to the use of fixed albedo parameters, and showed that incorporating more realistic bare ice albedo reduced the surface mass balance by approximately 145 Gt between 2000 and 2021. Although both studies focus on the GrIS, they differ in model structure and spatial resolution: MAR is a high-resolution regional climate model, while ELM-E3SM is part of a coarser-resolution global Earth system model. Despite these differences, both studies highlight a persistent bias—systematic overestimation of bare ice albedo. The convergence of evidence

from diverse modeling frameworks underscores the need to improve bare ice representation in land surface models. Building on these insights, our study examines the role of bare ice metamorphism, particularly changes in specific surface area, in driving progressive surface darkening. By isolating the feedback between evolving ice properties and surface energy balance, we propose a physically mechanism for the observed albedo decline. Our sensitivity analysis underscores how bare ice metamorphism can influence surface energy balance and the importance of incorporating such processes in future model developments.

**References:**

Antwerpen, R., Tedesco, M., Fettweis, X., Alexander, P.,and vandeBerg, W. J.: Assessing bare-ice albedo simulated byMAR overthe Greenland icesheet(2000–2021) andimplications formeltwater production estimates, The Cryosphere, 16(10), 4185–4199, https://doi.org/10. 5194/tc-16-4185-2022, 2022.

Whicker-Clarke, A., Antwerpen, R., Flanner, M. G., Schneider, A., Tedesco, M., and Zender, C. S.: The effect of physically based ice radiative processes on Greenland ice sheet albedo and surface mass balance in E3SM, J. Geophys. Res.-Atmos., 129, e2023JD040241, https://doi.org/10.1029/ 2023JD040241, 2024.

**Other comments:**

**Section 1:**

1. L63-65 The sentence is too long. Please, split in two parts.

2. L63: extend → extent

**Responses:** Thanks for your suggestions. The revised sentence is as follows: "Fluctuations in the snowline dictate the relative extent of dark bare ice versus brighter snow (Ryan et al., 2019). These directly influence GrIS surface melt through the exposure of bare ice (Antwerpen et al., 2022) and the processes that darken bare ice itself (Chevrollier et al., 2023)."

3. L64: Surface melt is also associated with a reduction of snowpack thickness and is not only due to bare ice exposure. However, I agree with the fact that surface melting over bare ice surfaces contributes to GrIS mass loss. This should be better explained.

Responses: We sincerely appreciate your insightful comment. You highlighted that GrIS surface melt involves not only the bare ice exposure but also the reduction of snowpack thickness. In the paragraph in question, our focus was on how surface albedo influences melt processes. Compared with snowpack thinning accompanied by snow grain metamorphism and growth, the transition from snow to bare ice causes a more substantial reduction in albedo.

4. L66: was → is

**Responses:** Thank you for your comment. We have revised "was" to "is".

5. L73: in → over

**Responses:** Thank you for your comment. We have revised "in" to "over".

6. L130 (and also L154 and L203): You mention that ice albedo is 0.80 and 0.55 for VIS and NIR spectra. These values correspond more to the albedo values for fresh snow than to the albedo values for bare ice.

**Responses:** Thank you for pointing this out. This issue was also addressed in the major comments, and we have already provided a detailed response there. In short, we have re-run the CoLM SNICAR-AD simulations using more realistic bare ice albedo values (0.60 for VIS, 0.40 for NIR), consistent with Figure 10. The updated results (Figs. R6 and R7) confirm that SNICAR-ADv4 still improves albedo estimation. All relevant sections have been revised accordingly.

7. L136: in ablation season à during the ablation season

**Responses:** Revised as suggested (in ablation season → during the ablation season)

8. L145: properties

**Responses:** Sorry for this mistake. We have fixed it.

**Section 2:**

1. L151: features enhancements à "includes improvements in the representation of…" sounds better?

**Responses:** Thank you for this helpful suggestion. We agree that "includes improvements in the representation of..." more clearly conveys the methodological advancements. We have revised the text accordingly: "The CoLM version 2024 (CoLM 2024) used in this study is based on the CoLM 2014 and includes improvements in the representation of surface energy, hydrology, biogeochemical cycles, and anthropogenic disturbance processes."

2. L152: What are the improvements in the anthropogenic disturbances processes? Which kind of processes are you referring to?

**Responses:** Thank you for pointing this out. The anthropogenic disturbance improvements in CoLM 2024 include: (i) a new reservoir module (Cama-Flood-based) enhancing regulated river flow simulation, (ii) the GPAM1 crop model simulating major crops' climate responses, and (iii) the Li fire scheme replacing GlobFIRM to better capture human-influenced wildfires (Li et al., 2019). These represent part of the anthropogenic disturbance updates in the CoLM 2024, and additional improvements are documented in the CoLM technical manual (http://172.16.102.100/colm/).


[Figure]

Figure R9 Spectral reflectance of snow (68°56′33″N, 42°27′16″W, red circle), bare ice (68°05′10″N, 48°01′23″W, blue diamond), dark ice (69°32′25″N, 50°26′56″W, green square), and bare soil (68°23′02″N, 53°48′13″W, brown triangle) and RGB color composite image band 1, 4, and 3 taken on 12 July 2012 derived from MODIS (Shimada et al., 2016).


[Figure]

Figure R10. (a–f) Spectral albedo as a function of wavelength, snow or ice density, and the ice volume fraction of air. Shading indicates the full range of clean snow or ice albedo as a function of snow grain or air bubble radius, and the spectral albedo for an ice grain/air bubble with an effective radius of 180 μm is indicated by the colored line. Panel (a) is a snow layer; all the other panels are ice layers. The radius ranges from 30 μm (the highest albedo curves) to 20000 μm (the lowest albedo curve; Whicker-Clarke et al. 2022).


$$RMSE = \sqrt{\frac{1}{n}\sum_{i=1}^{n}(\alpha_{MODIS,\,i} - \alpha_{model,\,i})^2}$$

where n=21 (years), $\alpha_{modis,i}$ and $\alpha_{modis,i}$ are the MODIS and modeled albedo values for year $i$, respectively.

The "linear trend" refers to the temporal trend in annual summer albedo for each grid cell, calculated using least-squares regression over the 2000–2020 period. Trends are expressed in units of albedo change per year ($\Delta\alpha$/year). We will clarify this in the main text (Section 3.2): "Furthermore, comparative analysis of the spatial distributions of correlation coefficients, root mean square errors (RMSE), and linear trends (Figs. S1-S3) reveals that CoLM-SNICAR-ADv4 outperforms CoLM-SNICAR-AD across all evaluation metrics. These metrics were derived from each grid cell by comparing the 21-year summer albedo time series (2000–2020) from model simulations and MODIS observations: correlation coefficients assess temporal agreement, RMSE quantifies deviation magnitudes, and linear trends (obtained via

least-squares regression) capture interannual albedo changes. The comprehensive spatial evaluation demonstrates consistent improvements in both the spatial pattern and quantitative representation." We have also updated the relevant figure captions in the Supplementary Material accordingly.

13. L384-387: Not clear. I suggest a new formulation: The decrease in the positive bias of CoLM SNICAR-ADv4 can also be clearly seen in the shortwave, visible and near-infrared albedo time series, with the area-weighted mean albedo of the GrIS bare ice regions steadily decreasing throughout the summer period from 2000 to 2020, compared with CoLM SNICAR-AD.

**Responses:** We sincerely appreciate your constructive suggestion to improve clarity. This sentence has been revised as you suggest.

14. L390: SNICAR-ADv4 enabled simulations à CoLM SNICAR-ADv4 simulations

**Responses:** Thanks for your comment. We have updated the "SNICAR-ADv4 enabled simulations" to the "CoLM SNICAR-ADv4 simulations".

15. L391 and L392: MCD43C4 à MCD43C3

**Responses:** Thank you for pointing out these mistakes. We have corrected them.

16. L410: from → compared to

**Responses:** Thanks for your comment. We have corrected it.

17. L425: has significantly reduced

**Responses:** Thanks for your comment. We have corrected it.

18. L440: This could be confirmed or infirmed with new diagnostics (e.g. surface mass balance and/or runoff à See Major comments)

**Responses:** Thanks for your suggestion. Since CoLM does not support SMB calculation, we analyzed snow water equivalent (SWE), as detailed in our earlier reply.

19. L443: northeast ablation zone: Rather northwestern and western?

**Responses:** Thank you for pointing out this mistake. It should indeed be the northwestern and western ablation zone. We have carefully revised this throughout the manuscript and verified all related geographical references.

20. L455: control experiment: this is the first time you use this term. Please explain what is your control experiment.

**Responses:** We appreciate this important clarification request. To avoid confusion, we have replaced the term "control experiment" with a more precise description in the revised text. The sentence now reads: "From Fig. 10f, the difference in BBA shows a strong positive correlation with the specific surface area, with a correlation coefficient of 0.88 (significant at the 99% confidence level), since the two simulations differ solely in their prescribed bare ice physical properties in the land surface model."

21. L456: commence → starts/begins

**Responses:** Thank you for your comment. We have revised "commence" to "start" in the revised manuscript.

22. L465: region → regional

**Responses:** Thank you for pointing out this mistake. We have corrected "region" to "regional" in the revised manuscript.

23. L468: effect → affect

**Responses:** Thank you for pointing out this mistake. We have corrected "effect" to "affect" in the revised manuscript.

24. L476: speciafic →specific

**Responses:** Thank you for pointing out this typo. We have corrected "speciafic" to "specific" in the revised manuscript.

25. L489: 2021 → 2020

**Responses:** Thank you for pointing out this mistake. We have corrected "2021" to "2020" in the revised manuscript.

26. L492-493: strong climate response: This sounds like an overstatement

**Responses:** We appreciate the your valid concern about terminology precision. The original phrasing has been revised to: "...suggesting that even a slight reduction in bare ice albedo can produce noticeable climate responses in ablation region."

27. L497: in ablation zone → in the ablation zone

**Responses:** Thank you for pointing out this mistake. We have corrected it.

28. L513-514: impact of the glacier calving (dynamic process) and submarine melting: Add a reference and/or develop your arguments.

**Responses:** Thanks for your comment. In the revised manuscript, we have refocused the discussion exclusively on sea-level linkages, as the glacier-specific connections were indeed tenuous. The text now states: "Such feedback is projected to amplify the GrIS's contribution to global sea level rise by enhancing both surface melting and runoff generation (Ryan et al., 2019). The potential acceleration of these sea-level-relevant processes underscores a critical research priority for improving future projections."

**Responses:** Thanks for your constructive suggestion. The color schemes in these figures have been systematically updated to employ higher-saturation, perceptually optimized palettes.

---

## Author Response (AR2)

**Responses to reviewer #1's comments about the manuscript, entitled "Enhanced MODIS-derived ice physical properties within CoLM revealing bare ice-snow-albedo feedback over Greenland" (EGUSPHERE -2025-230)**

We would like to sincerely thank reviewer #1 for their thoughtful and constructive comments, which have greatly improved the quality and clarity of our manuscript. We have carefully addressed each of the comments and revised the manuscript accordingly. Our responses to the individual points are detailed below.

The revisions include clearer explanations of the glacier and snow-related processes represented in CoLM, providing more accurate context for the model's capabilities and limitations, as well as the addition of a synthetic description of the physical processes underlying bare ice metamorphism.. We also improved the description of evaluation metrics used to compare modeled and observed albedo, and expanded the discussion of key figures such as Figure 10e.

Additionally, during the revision process, we adopted a helpful suggestion from Reviewer #2 to move Figure 5 (previously illustrating the parameter sensitivity of ice spectral albedo in SNICAR-ADv4) ahead of Figure 3. Accordingly, we now present this content in a standalone subsection titled 2.3 Parameter sensitivity of ice spectral albedo in SNICAR-ADv4. This adjustment led to changes in the numbering of several figures, which we kindly note here for ease of reference.

The red text in this response file highlights the specific changes we made to the manuscript. Once again, we appreciate the reviewer's time and valuable input, which have been instrumental in strengthening this work.

**Specific comments**

1. In my first review, I suggested to evaluate the impact of bare ice metamorphism on the surface mass balance and the runoff. According to the authors, the computation of SMB and runoff (from snow melt and ice melt) is not possible because the glacier scheme implemented within the CoLM framework assumes fixed ice thickness and mass. However, as I understand so far, all the elements needed to calculate an energy balance are present in the model and could be used to calculate the amount of ice that could be melted. I would greatly appreciate that the authors provide a detailed response about that point. What would be even more convincing would be to calculate

the amount of ice that could be melted based on the energy available. This would allow to compute runoff and SMB.

**Responses:** We sincerely appreciate the reviewer's insightful suggestion regarding the estimation of SMB and runoff based on available melt energy. CoLM2024 indeed computes a comprehensive surface energy balance, incorporating all relevant fluxes as well as snow-related phase change processes such as melting, sublimation, and refreezing. This detailed representation allows for a robust estimation of melt potential within the snowpack. However, the treatment of glacier ice within CoLM remains relatively simplified. The current glacier scheme assumes a fixed ice thickness and non-evolving ice mass, meaning that while the ice column is thermodynamically active, it does not dynamically adjust to melt by reducing ice thickness or contributing to mass loss, and lacks a representation of the snow-to-ice transition. Consequently, the model cannot convert melt energy directly into actual ice mass depletion or runoff generation, and prognostic SMB diagnostics are not available in the current framework.

Nevertheless, we agree that a diagnostic estimate of melt potential based on residual energy (i.e., energy available after meeting other demands like warming and sublimation) is feasible. Such estimates would reflect theoretical melt amounts, but would not be routed through the model's hydrological system or affect the surface mass budget. Therefore, although diagnostic runoff can be calculated, it would not reflect realistic mass loss or runoff processes under the current model structure.

To clarify this limitation, we have revised the manuscript to explicitly distinguish between diagnostic melt potential and physically consistent runoff generation. We have also emphasized that extending CoLM with a coupled glacier mass balance and hydrology scheme would be a valuable direction for future work.

In any case I recommend clearly describing what the glacier module implemented in CoLM is capable of representing and what it cannot. In the same way, I would appreciate that the processes simulated by the snow model were mentioned. Figure 1 does not illustrate the processes which are represented in snow and ice modules.

**Responses:** We thank the reviewer for this constructive and helpful comment. To improve clarity, we have added a new paragraph in Section 2.1 Snow and Ice Albedo Schemes to clearly describe the capabilities and limitations of the glacier and snow modules implemented in CoLM.

Specifically, the glacier module in CoLM employs a simplified representation of land ice. It does not simulate SMB components, dynamic glacier geometry, or meltwater runoff from glacier ice. Glacier thickness is fixed, with no treatment of snow-to-ice transition, ice meltwater production, or sublimation from glacier surfaces. In addition, each glacier grid cell is represented as a single elevation unit, without accounting for subgrid elevation variability. These simplifications distinguish CoLM from more advanced land ice schemes that incorporate elevation classes, explicitly compute SMB terms, and allow for dynamic ice evolution and runoff routing.

Despite these limitations, CoLM does simulate several key surface and subsurface processes relevant to snow and ice physics: (1) It resolves full surface energy balance components, including radiative, turbulent, and ground heat fluxes for ice and snow. (2) It simulates vertical temperature evolution using a multilayer structure (5 snow layers and 10 glacier ice layers). (3) It includes snow hydrological processes such as melt, percolation, retention, refreezing, and runoff generation from snowmelt. (4) It accounts for albedo evolution driven by snow aging and explicitly simulates ice optical properties via the SNICAR-ADv4 radiative transfer scheme. (5) It allows for sublimation from snow-covered surfaces, although sublimation from glacier ice is not represented due to the absence of ice-atmosphere mass exchange mechanisms.

These structural differences help explain the counterintuitive model behavior observed in our simulations. Specifically, when albedo decreases, the resulting increase in absorbed energy does not lead to enhanced runoff as expected, because CoLM lacks explicit representations of mass evolution and meltwater discharge from bare ice. Instead, the excess energy is unrealistically retained within the system and partially offset by increased sublimation. To clarify these limitations, we have added a new paragraph in Section 2.1 (Snow and ice albedo schemes) that distinguishes the physical processes represented in the snow and glacier components. The paragraph reads as follows:

In addition to the albedo scheme, we briefly describe the physical processes represented by the glacier and snow modules in CoLM to clarify model capabilities. The glacier component is designed to capture essential surface thermodynamic processes, including full surface energy balance calculations and subsurface heat diffusion through a multi-layer ice column. However, it omits several key elements found in more advanced land ice schemes: (1) the model assumes fixed ice thickness and does not track accumulation or ablation, lacking mass-conserving SMB

computation; (2) glacier geometry is static, with no coupling to an ice sheet model for dynamic evolution; and (3) meltwater generated from glacier ice is retained rather than routed to runoff, leading to unrealistic surface water storage. In contrast, the snow component in CoLM simulates several critical processes: (1) multi-layer snowpack energy and mass balance, including radiative, turbulent, and conductive heat fluxes; (2) vertical snow temperature evolution and phase changes; (3) melt, liquid water percolation, refreezing, sublimation and snowmelt runoff generation; and (4) snow aging and albedo evolution, with consideration of the impacts of LAPs, as represented by SNICAR-AD/SNICAR-ADv4.

To be honest, I am not convinced that the bare ice metamorphism has a significant impact on the SWE. As far as I am concerned, I often saw in the literature orders of magnitude for the SWE reaching several hundreds of mm (see for example Krinner et al., 2018) hence 1.345 mm represents barely 1% of the orders of magnitude generally encountered. In the same way a change in temperature forcing does not seem significant (as I raised in my first review). I acknowledge that accounting for bare ice metamorphism could improve the representation of surface processes, but I recommend to be more cautious and avoid overstatements in that paper.

**Responses:** We thank the reviewer for this valuable comment. We agree that the change in SWE (~1.345 mm) and temperature response are modest in magnitude and do not represent a large impact at the ice sheet scale. To address this concern, we have revised the manuscript to avoid the use of terms like "significant" and instead describe the effects as "detectable" or "measurable." That said, our aim was not to emphasize the magnitude of the SWE change itself, but to illustrate that even relatively small changes in bare ice albedo ($\Delta\alpha$ = 0.032) can propagate into measurable changes in near-surface temperature (0.071 °C) and snow cover extent (−0.011). The revised text now reflects a more restrained interpretation.

2. I also think that that a synthetic description of physical processes underlying bare ice metamorphism is lacking. Explain, for example, how and why the size of air bubbles can change (under which influences?). What is the link between the radius or the volume of air bubbles and the physical properties of ice (e.g. density)? What role do bubbles play in the melting and refreezing processes of ice?

**Responses:** We thank the reviewer for raising this important point. In the revised manuscript (Section 2.2 Parameter sensitivity of ice spectral albedo in SNICAR-ADv4; originally Section 2.3), we have added a synthetic description of the

physical processes underlying bare ice metamorphism following the sensitivity experiments using the standalone SNICAR-ADv4 with varying input parameters. While the preceding analysis illustrates how physical properties of bare ice (e.g., density and bubble radius) affect spectral albedo, the new addition explains how these properties are shaped by natural atmospheric and radiative conditions, with particular emphasis on the role of air bubbles and their connection to specific surface area (SSA) and ice density during melting and refreezing processes.

As for how and why the size of air bubbles can change, the metamorphism of bare ice is primarily driven by differential solar absorption along grain boundaries and bubble surfaces, as described by Müller and Keeler (1969). Under clear-sky conditions, penetrating shortwave radiation induces subsurface melting, which dissolves the ice matrix and expands air bubble volume. This process can be described by $V_{air} = (\rho_{ice} - \rho_{layer})/\rho_{ice}$ where $\rho_{ice} = 917$ kg/m³. As bubble volume increases and $\rho_{layer}$ decreases, the effective bubble radius grows, thereby increasing the specific surface area (SSA; SSA $= 3V_{air}/(\rho_{layer} \cdot r_{eff})$). This microstructural transition forms a porous, low-density "weathering crust," commonly observed in ablation zones (Müller and Keeler, 1969). Meteorological conditions modulate these processes. Under overcast, windy, and warm conditions, this crust is preferentially removed, exposing denser, glazed ice beneath. Temperature-driven grain sintering and densification further reduce SSA by smoothing and coalescing ice grains (Flanner and Zender, 2006; Hofer et al., 2017). Concurrently, air bubble growth from differential solar heating and subsurface melting continues to modify the microstructure and optical properties of the ice.

Regarding the link between air bubbles and the physical properties of ice, microstructural variations in the uppermost ice layer play a critical role in modulating the surface energy balance (Jonsell et al., 2003). The growth of numerous small air bubbles increases SSA and enhances light scattering, which can transiently elevate surface albedo. This effect is particularly pronounced in the visible spectrum (Whicker-Clarke et al., 2022). Conversely, when meltwater infiltrates and subsequently refreezes within the pore space, the bubble structure collapses, producing denser, low-SSA glaze ice with reduced reflectivity. These microstructural transitions help explain the short-term variability in surface optical and thermal properties.

As for the role of air bubbles in the melting and refreezing processes of ice, they enhance subsurface solar absorption by concentrating energy along bubble–ice

interfaces (Light et al., 2004), promoting localized melting and the expansion of pore spaces. This increases the SSA and lowers thermal conductivity, helping to sustain subsurface warming (Brandt and Warren, 1993). During cooler or windy periods, meltwater that has infiltrated the porous structure can refreeze, compacting the ice matrix and reducing porosity. This leads to the formation of denser, low-SSA glaze ice with diminished albedo. In this way, air bubbles play a critical role in regulating both energy absorption and phase transitions, enabling rapid shifts in the surface's optical and thermal properties in response to meteorological forcing.

The paragraph to be added to the manuscript is as follows:

While these controlled simulations clarify the fundamental optical behavior of ice under idealized conditions, natural environments involve more complex interactions shaped by microstructural evolution and meteorological forcing. A synthetic description of bare ice metamorphism includes the evolution of air bubbles and density: newly fallen snow starts with low density and open pore spaces, which become compacted through wind-driven grain fragmentation and rounding, forming wind slabs. Further densification occurs via grain-boundary sliding and pressure-induced deformation, during which air bubbles become sealed and gradually shrink under compression (Tedesco et al., 2016). In ablation zones, these densification processes are coupled with surface metamorphism. Glaciers undergoing melt often develop a porous weathering crust composed of loosely interlocked crystals, formed by differential solar absorption along grain boundaries, a process termed internal ablation (Müller and Keeler, 1969). Under overcast, windy, and warm conditions, this crust is preferentially removed, exposing denser, glazed ice beneath. Temperature-driven grain sintering and densification further reduce SSA by smoothing and coalescing ice grains (Flanner and Zender, 2006; Hofer et al., 2017). Concurrently, air bubble growth from differential solar heating and subsurface melting continues to modify the microstructure and optical properties of the ice.

**References:**

Arnaud, L., Barnola, J. M., & Duval, P. (2000). Physical modeling of the densification of snow/firn and ice in the upper part of polar ice sheets. In T. Hondoh (Ed.), Physics of Ice Core Records (pp. 285–305). Hokkaido University Press.

Brandt, R. E., & Warren, S. G. (1993). Solar-heating rates and temperature profiles in Antarctic snow and ice. Journal of Glaciology, 39(131), 99–110. https://doi.org/10.3189/S0022143000015913

Flanner, M. G., Zender, C. S., Randerson, J. T., & Rasch, P. J. (2007). Present-day climate forcing and response from black carbon in snow. Journal of Geophysical Research: Atmospheres, 112(D11), D11202. https://doi.org/10.1029/2006JD008003

Hofer, S., Tedstone, A. J., Fettweis, X., & Bamber, J. L. (2017). Decreasing cloud cover drives the recent mass loss on the Greenland Ice Sheet. Science Advances, 3(6), e1700584. https://doi.org/10.1126/sciadv.1700584

Holmgren, B. (1971). Climate and energy exchange on a sub-polar ice cap in summer: Arctic Institute of North America Devon Island Expedition 1961–1963, Part E. Radiation climate (Meddelande 111). Uppsala Universitet, Meteorologiska Institutionen.

Jonsell, U., Hock, R., & Holmgren, B. (2003). Spatial and temporal variations in albedo on Storglaciären, Sweden. Journal of Glaciology, 49(164), 59–68. https://doi.org/10.3189/172756503781830980

Lewis, G., Osterberg, E., Hawley, R., Marshall, H. P., Meehan, T., Graeter, K., et al. (2021). Atmospheric blocking drives recent albedo change across the western Greenland Ice Sheet percolation zone. Geophysical Research Letters, 48(14), e2021GL092814. https://doi.org/10.1029/2021GL092814

Light, B., Maykut, G. A., & Grenfell, T. C. (2004). A two-dimensional Monte Carlo model of radiative transfer in sea ice. Journal of Geophysical Research: Oceans, 109(C2), C02019. https://doi.org/10.1029/2003JC001956

Müller, F., & Keeler, C. M. (1969). Errors in short term ablation measurements on melting ice surfaces. Journal of Glaciology, 8(52), 91–105. https://doi.org/10.3189/S0022143000031215

Tedesco, M., Doherty, S., Fettweis, X., Alexander, P., Jeyaratnam, J., & Stroeve, J. (2016). The darkening of the Greenland ice sheet: Trends, drivers, and projections (1981–2100). The Cryosphere, 10, 477–496. https://doi.org/10.5194/tc-10-477-2016

Tedstone, A. J., Bamber, J. L., Cook, J. M., Williamson, C. J., Fettweis, X., Hodson, A. J., & Tranter, M. (2017). Dark ice dynamics of the south-west Greenland Ice Sheet. The Cryosphere, 11, 2491–2506. https://doi.org/10.5194/tc-11-2491-2017

Whicker-Clarke, A., Flanner, M. G., Dang, C., Zender, C. S., Cook, J. M., and Gardner, A. S.: SNICAR-ADv4: A physically based radiative transfer model to represent the spectral albedo of glacier ice, The Cryosphere, 16(4), 1197–1220,

3. There are also confusions in the Discussion Section. First, you mention that in ablation areas dynamical processes dominate mass loss (Lines 598-599). This is not

true: recent observations show that surface processes are responsible for about 2/3 of the total mass loss in Greenland (see for example Ryan et al., 2019, van den Broeke et al. 2016, The IMBIE Team, 2020). By the way, this is what you explain in the Introduction (Lines 48-51). Second, the use of a dynamic ice sheet model will not help to properly compute SMB and runoff (Lines 604-605). An ice-sheet model is generally forced by surface climate or by SMB and simulated the dynamics of an ice sheet. SMB can be provided by a combined snow-ice module able to compute runoff, sublimation and refreezing.

**Responses:** We thank the reviewer for the helpful clarification. We have revised the Discussion to clearly state that surface processes (e.g., meltwater runoff, sublimation, refreezing) dominate Greenland's total mass loss, accounting for about two-thirds. Accordingly, Lines 598–599 were updated to: "Although computationally efficient, this simplification systematically underestimates meltwater export from Greenland's ablation zones, where surface processes and especially meltwater runoff are the dominant contributors to mass loss (Ryan et al., 2019, van den Broeke et al., 2016, The IMBIE Team, 2020)."

Regarding SMB and runoff modeling, we agree that a dynamic ice sheet model alone is insufficient for physically consistent SMB estimation. Accurate SMB requires a physically based land surface or snow-ice model resolving key processes such as melt, runoff generation, sublimation, refreezing, and snow-to-ice transformation, ideally with elevation-aware forcing. Compared with CLM5, CoLM lacks snow-to-ice transformation, retains all meltwater without runoff, and does not include elevation-based forcing, limiting its SMB and runoff accuracy. We revised Lines 604–605 to: "…improving the representation of surface mass balance and ice-melt runoff through a more complete snow and ice surface scheme…"

Our study represents a first step toward improving the surface radiative representation in the CoLM by introducing SNICAR-ADv4, which better captures the bare ice albedo feedback. This improvement is a prerequisite for future development of a more comprehensive SMB modeling framework within the CoLM.

**References:**

Ryan, J.V., Smith, L. C., van As, D., Cooley, S. W., Cooper, M. G., Pitcher, L. H., and Hubbard, A.: Greenland Ice Sheet surface melt amplified by snowline migration and bare ice exposure, Science Advances, 5, eaav3738, https://doi.org/10.1126/sciadv.aav3738, 2019

The IMBIE team: Mass balance of the Greenland ice sheet from 1992 to 2018, Nature, 579, 233-239, https://doi.org/10.1038/s41586-019-1855-2, 2020.

van den Broeke, M., Enderlin, E. M., Howat, I. M., Kuipers Munnneke, P., Noël, B. P. Y., van de Berg, W. J., van Meijgaard, E., Wouters, B.: On the recent contribution of the Greenland ice sheet to sea level change, The Cryosphere, 10, 1933-1946, https://doi.org/10.5194/tc-10-1933-2016, 2016

4. I don't really understand why you choose 650 kg m$^{-3}$ for the lower bound of ice density. In general, the appearance of ice occurs for density values around 750-850 kg m-3. Your choice should be justified and commented in the Discussion Section. In particular, what would be the impacts on your results if you have chosen a more realistic value for the lower boundary of ice density?

**Responses:** Thank you for raising this important point. We agree that the selection of the density threshold as the lower bound of ice density is a key parameter that requires careful justification. We apologize for not explaining this more clearly in our previous response.

In our study, we adopted 650 kg m$^{-3}$ as the lower bound of ice density based on the recommendations from the SNICAR-ADv4 model developers (Whicker-Clarke et al., 2022). As stated in the SNICAR-ADv4 documentation and supported by relevant studies (Bender et al., 1997; Dadic et al., 2013), SNICAR-ADv4 shows better agreement with observed spectral albedos when media with densities above 650–700 kg m$^{-3}$ are treated as ice layers (Fig. R1b). This choice reflects a compromise between physical realism and radiative consistency in the model. While ice formation generally occurs at higher densities (~830 kg m$^{-3}$), the optical properties of firn with densities above 650 kg m$^{-3}$ tend to align more closely with those of ice rather than snow in terms of radiative transfer, especially when air bubbles become dominant scatterers within a solid matrix.

Regarding the practical implications of threshold selection, we emphasize that adopting a higher density threshold (750–850 kg m$^{-3}$) would systematically impact our retrieval results in two key ways: (1) In standalone SNICAR-ADv4 simulations, lower-density ice layers (650–750 kg m$^{-3}$) generally produce higher Band 2 albedo values; and (2) Using a higher threshold would narrow the range of our Band 2 albedo–ice property lookup table. This limitation becomes particularly critical when processing MCD43C3 observations, as higher observed Band 2 albedos may exceed the bounds of the restricted lookup table, thereby preventing the retrieval of

associated ice properties.

Therefore, while we acknowledge that 650 kg m⁻³ somewhat underestimates the typical physical transition to glacier ice, we selected this threshold to balance physical realism with the practical need for comprehensive retrieval coverage. This compromise ensures that our methodology remains both physically justifiable and operationally robust across the full range of observed albedo values.

[Figure]

Figure R1. Spectral albedo measured by Dadic et al. (2013a) compared to SNICAR-ADv4 modeled spectral albedo. The model parameters are tightly constrained with quantitative measurements made by Dadic et al. (2013a). Figure reproduced from Whicker-Clarke et al. (2022), included here for review purposes only.

**References:**

Whicker-Clarke, A., Flanner, M. G., Dang, C., Zender, C. S., Cook, J. M., and Gardner, A. S.: SNICAR-ADv4: A physically based radiative transfer model to represent the spectral albedo of glacier ice, The Cryosphere, 16(4), 1197–1220, https://doi.org/10.5194/tc-16-1197-2022, 2022.

Bender, M., Sowers, T., and Brook, E.: Gases in ice cores, P. Natl. Acad. Sci., 94, 8343–8349, https://doi.org/10.1073/pnas.94.16.8343, 1997.

Dadic, R., Mullen, P. C., Schneebeli, M., Brandt, R. E., and Warren, S. G.: Effects of bubbles, cracks, and volcanic tephra on the spectral albedo of bare ice near the Transantarctic Mountains: Implications for sea glaciers on Snowball Earth, J. Geophys. Res.-Earth, 118, 1658–1676, https://doi.org/10.1002/jgrf.20098, 2013.

**Minor comments:**

1. Title → MODIS

**Responses:** Thank you for your comment. We have revised "MOIDS" to "MODIS".

2. Line 21: In the Common Land Model

**Responses:** Thank you for your comment. We have fixed it.

3. Line 21: to represent

**Responses:** Thank you for your comment. We have fixed it.

4. Line 26: 0.053 does not seem fully negligible. Maybe it would be better to remove "only"

**Responses:** Thank you for your comment. We have removed "only".

5. Line 41: The negative mass balance → The decreasing mass balance

**Responses:** Thank you for your comment. The text has been modified to use "decreasing mass balance" as suggested.

6. Line 48: frontal → dynamical

**Responses:** Revised as suggested.

7. Lines 44-46: To be shifted at the end of the paragraph.

**Responses:** Thank you for the suggestion. We have moved the sentence to the end of the paragraph as requested, and revised it to: "These melting processes are driven by a combination of factors, including atmospheric warming, a reduced water retention capacity of firn due to densification and a lower surface albedo (Hofer et al., 2017; King et al., 2020; Ryan et al., 2024)."

8. Line 45 : Explain why there is a reduction of the water retention capacity

**Responses:** Thank you for your comment. We have now clarified in the manuscript that the reduced water retention capacity of firn is primarily due to three processes identified in recent studies (Hofer et al., 2017; King et al., 2020): (1) the formation of impermeable ice layers through refreezing of meltwater within the firn, which limits percolation and enhances surface runoff; (2) progressive firn densification and decreasing porosity caused by repeated melt–refreeze cycles, reducing available pore space; and (3) rising firn temperatures due to atmospheric warming and latent heat release during refreezing, which further accelerates densification. These changes

collectively diminish the firn's buffering capacity and increase meltwater runoff to the ocean.

**References:**

Hofer, S., Tedstone, A. J., Fettweis, X. and Bamber, J. L.: Decreasing cloud cover drives the recent mass loss on the Greenland Ice Sheet, Sci. Adv., 3, e1700584, https://doi.org/10.1126/sciadv.1700584, 2017.

King, M. D., Howat, I. M., Candela, S. G., Noh, M. J., Jeong, S., Noël, B. P. Y., Van den Broeke, M. R., Wouters, B., and Negrete, A.: Dynamic ice loss from the Greenland Ice Sheet driven by sustained glacier retreat, Commun. Earth Environ., 1(1), 1. https://doi.org/10.1038/s43247‐020‐0001‐2, 2020.

9. Line 114: Ice → ice

**Responses:** Thank you for your comment. We have fixed it.

10. Line 117: This is not a parametrization as albedo values are prescribed to constant values.

**Responses:** Thank you for the comment. We revised the sentence to clarify that prescribing constant albedo values is not a parameterization, and revised it to: "Prescribing constant albedo values does not represent the physical variability of solid ice albedo or its spectral changes under varying conditions."

11. Line 122: Explain what the default ELM method is

**Responses:** Thank you for the comment. The default ELM method refers to the prescribed constant albedo values of 0.6 in the visible and 0.4 in the NIR bands in the Energy Exascale Earth System Model (E3SM), as stated earlier in the paragraph.

12. Line 137: during the ablation season

**Responses:** Thank you for your comment. We have fixed it.

13. Line 150: → schemes

**Responses:** Thank you for your comment. We have fixed it.

14. Line 202: Add a reference for the mean altitude of the equilibrium line.

**Responses:** Thank you for your comment. Thank you for your comment. We have added the reference Antwerpen et al., 2022 to support the statement on the mean altitude of the equilibrium line.

**References:**

Antwerpen, R., Tedesco, M., Fettweis, X., Alexander, P.,and vandeBerg, W. J.: Assessing bare-ice albedo simulated byMAR overthe Greenland icesheet(2000–2021) andimplications formeltwater production estimates, The Cryosphere, 16(10), 4185–4199, https://doi.org/10. 5194/tc-16-4185-2022, 2022.

15.  Line 257: This information is not relevant as cryosphere is defined as any form of frozen water at the Earth surface (whatever the density).

**Responses:** Thank you for pointing this out. We agree that our original phrasing was misleading and could be interpreted as a definition of cryospheric media. We have revised the sentence to clarify that our intention was merely to define a density threshold for ice within the model, rather than to provide a physical definition of the cryosphere. The revised sentence now reads: "Following the SNICAR-ADv4 modeling configuration, ice with densities above $650\,\mathrm{kg\cdot m^{-3}}$ is treated as bubbly ice, following the modeling approach in Whicker-Clarke et al. (2022), which showed optimal agreement with in situ measurements."

**References:**

Whicker-Clarke, A., Flanner, M. G., Dang, C., Zender, C. S., Cook, J. M., and Gardner, A. S.: SNICAR-ADv4: A physically based radiative transfer model to represent the spectral albedo of glacier ice, The Cryosphere, 16(4), 1197–1220, https://doi.org/10.5194/tc-16-1197-2022, 2022.

16.  Line 261: corresponding → correspond

**Responses:** Thank you for your comment. We have fixed it.

17. Line 269: alpha is often used to denote albedo. Maybe you could find another notation

**Responses:** We appreciate your comment and have revised the notation accordingly by replacing α with SSA in the equation to improve clarity.

18. Line 272: I don't understand what you mean with "functional degeneracy". Please, find another expression

**Responses:** Thank you for pointing this out. We agree that the term "functional degeneracy" was unclear. We have revised the sentence for clarity as follows: "Figure 3b shows the band 2 albedo from the SNICAR-ADv4 lookup table as a function of

SSA, illustrating that the modeled albedo is primarily determined by SSA rather than the specific combination of ice density and bubble size."

19. Line 289: Problem with the figure reference

**Responses:** Thank you for bringing this to our attention. The figure reference has been corrected from "Figs. R3d and 2e" to "Figs. 4d and 4e" in accordance with the revised figure numbering (originally Figure 3).

20. Line 323: within the ice

**Responses:** Thank you for your comment. We have added "the" as suggested, and the phrase now reads "within the ice."

21. Line 364: Pb with figure reference

**Responses:** Thank you for pointing this out. We have corrected the figure reference from "Figures 3b–e" to "Figures 4c–f" following the updated figure numbering in the revised manuscript.

22. Line 368: Please refer to Equations 1 and 2

**Responses:** Thank you for the suggestion. We have revised the sentence to explicitly refer to Equations 1 and 2 for clarity, and corrected the figure references accordingly. The revised sentence now reads: "The bare ice density gradually decreases from the lower-elevation coastal regions toward the interior, while the volume fraction of air shows an opposite pattern, as it is calculated from the bulk ice-air mixture density and the density of pure ice (Figs. 4d and f; Eqs. 1 and 2)."

23. Line 369: represents

**Responses:** Thank you for your comment. We have corrected "represent" to "represents".

24. Line 376: There is no Figure 3g

**Responses:** Thank you for pointing this out. The correct reference is Figure 5g (originally Figure 4g), and this has been corrected in the revised manuscript.

25. Line 380 but is also influenced

**Responses:** Thank you for your comment. We have fixed it.

26. Line 381-382 and Figure 6a: I don't understand what you mean because in the

interior of the GrIS, land ice underlying the snowpack is the only land cover type. In the figure, land ice cover is zero in the GrIS interior. Please clarify.

**Responses:** Thank you for your comment. We apologize for the insufficient explanation regarding Figure 6a. The map in Figure 6a shows the land ice area with permanent snow cover removed. In the interior of the GrIS, the surface is continuously covered by permanent snow, so this region is not counted as land ice in Figure 6a. This figure only considers the underlying surface type. In contrast, Figure 6b further accounts for snow cover on top of the land ice and presents the bare ice extent. We have clarified this distinction in the revised manuscript to avoid confusion.

27. Figure 6c: This is not clear. Does it mean for example that ~14-15% of the grid cells have bare ice fractions between 0 and 0.5? Please clarify both in the main text and in the figure caption.

**Responses:** We thank the reviewer for the helpful comment. We have clarified both the main text and the figure caption accordingly. Specifically, Figure 6c shows the relative frequency distribution of the mean exposed bare ice fraction, calculated across grid cells that exhibit nonzero exposed bare ice during JJA from 2000 to 2020. Grid cells with no exposed bare ice throughout the study period were excluded from this analysis. Each bar represents the proportion of these remaining grid cells whose mean bare ice fraction falls within a given interval, relative to the total number of grid cells with some exposed bare ice. For example, approximately 14% of the bare ice grid cells have a mean exposed bare ice fraction between 0 and 0.05, and about 8% fall between 0.05 and 0.1. This interpretation has been clarified in both the revised figure caption and the corresponding section of the main text.

[Figure]

Figure R2. Spatial distribution of (a) the fraction of land ice underlying the snowpack, excluding interior regions of the GrIS that remain fully snow-covered throughout JJA (2000–2020), and (b) the mean exposed bare ice fraction during JJA over the same period, based on snow cover simulated by CoLM using the SNICAR-ADv4 scheme. Panel (c) shows the relative frequency distribution of mean exposed bare ice fraction, considering only grid cells with nonzero bare ice exposure. Each bar indicates the percentage of these grid cells whose mean bare ice fraction falls within a given interval, relative to the total number of bare ice grid cells.

28. Lines 401-402: What is the default albedo scheme

**Responses:** We thank the reviewer for pointing this out. By "default albedo scheme," we refer to the standard configuration shared by both SNICAR-AD and SNICAR-ADv4 in our simulations. Specifically, this includes the use of spherical snow grain shape, the adding-doubling radiative transfer solver, and the external mixing state for black carbon (BC) and dust in snow. We have revised the text for clarity as follows: "Both SNICAR-AD and SNICAR-ADv4 simulations use the same default snow albedo configuration, which includes spherical snow grains, the adding-doubling radiative transfer solver, and external mixing of BC/dust with snow."

29. Line 403: Which albedo are you referring to?

**Responses:** Thank you for your comment. We apologize for the ambiguity. In this sentence, we are referring specifically to the ice albedo simulated by the two schemes. We have revised the sentence for clarity: "In other words, the differences in simulated albedo between SNICAR-AD and SNICAR-ADv4 arise solely from their different treatments of ice albedo, as the snow albedo configuration remains identical."

30. Lines 404 and 409: Problems with figure references

**Responses:** We thank the reviewer for pointing this out. We have corrected the figure references accordingly. Specifically, the figures related to CoLM-SNICAR-AD have been revised from the original Figs. 7d–f to Figs. 7a–c, and the figures related to CoLM-SNICAR-ADv4 have been revised from Figs. 7d–i to Figs. 7d–f in the revised manuscript.

31. Line 406: regions

**Responses:** Thank you for your comment. We have fixed it.

32. Figure 9: How do you explain that uncertainties are always larger with

SNICAR-ADv4 compared to SNICAR-AD? (same for MODIS)

**Responses:** We thank the reviewer for the thoughtful question. We have clarified this point in the revised manuscript. The larger uncertainties (i.e., double standard error) observed in both SNICAR-ADv4 and MODIS albedo across different bare ice fraction bins indicate a higher degree of spatial heterogeneity in albedo values. This reflects the fact that both SNICAR-ADv4 and MODIS better capture the spatial variability of surface albedo over varying bare ice conditions compared to SNICAR-AD. Therefore, the increased uncertainty in SNICAR-ADv4 is not a drawback, but rather suggests that it more realistically represents the natural variability seen in satellite observations.

33. Line 414: The correlation coefficients do not assess the temporal agreement between two time series.

**Responses:** We thank the reviewer for this important clarification. We agree that correlation coefficients primarily measure the linear relationship between time series rather than absolute temporal agreement. The text will be revised to more precisely state: "Correlation coefficients evaluate the phase similarity of interannual variations, RMSE quantifies deviation magnitudes, and linear trends capture systematic changes."

34. Line 415: Here, RMSE is the spatial RMSE as your sum is over the 21 years of the analysis. This should be mentioned in the main text as well as in the Supplement Figures S1-S3 should be commented. Note that crosses (see figure caption for S1 and S3) are not visible.

**Responses:** We thank the reviewer for the helpful comment. We have clarified in both the main text (Line 415) and the Supplement that the RMSE shown in Figures S1–S3 is computed as the spatial RMSE, based on the 21-year summer (JJA) time series at each grid cell. We have revised the corresponding text in the manuscript to:

Furthermore, for each grid cell over the GrIS bare ice region, we computed the root-mean-square error (RMSE) between the MODIS observed albedo and model-simulated albedo (CoLM-SNICAR-AD/SNICAR-ADv4) time series (2000-2020, 21 summer values per cell). Comparative analysis of the spatial distributions of correlation coefficients, RMSE, and linear trends (Figs. S1-S3) reveals that CoLM-SNICAR-ADv4 outperforms CoLM-SNICAR-AD across all evaluation metrics. These metrics were derived by comparing the 21-year summer albedo time series from model simulations and MODIS observations at each grid cell:

correlation coefficients evaluate the phase similarity of interannual variations, RMSE quantifies deviation magnitudes, and linear trends (obtained via least-squares regression) capture interannual albedo changes. The comprehensive spatial evaluation demonstrates consistent improvements in both the spatial pattern and quantitative representation.

Regarding the visibility of crosses in Figures S1 and S3: for Figure S1, the significance markers (crosses) were difficult to see due to the darker background color and image compression. We have slightly lightened the colormap and adjusted the figure export settings to prevent compression of the original high-resolution image, thereby improving the visibility of the crosses. For Figure S3, crosses are only shown where linear trends are statistically significant ($p < 0.05$). Since only a few grid cells meet this criterion, the scarcity of visible crosses reflects the limited spatial extent of significant trends. This has been clarified in the revised figure caption.

35. Line 484: zones

**Responses:** Thank you for your comment. We have fixed it.

36. Figure10 is not fully commented. Please provide also further details in the main text.

**Responses:** We appreciate the reviewer's comment regarding the insufficient description of Figure 10, especially panel (e). We have now added a more detailed explanation of Figure 10e in the main text. "Although the mean differences in 2-m air temperature, snow cover, and snow water equivalent appear small, there are a considerable number of grid cells showing substantially higher 2-m air temperature differences and notably lower snow cover and snow water equivalent values. This indicates that certain regions of the GrIS exhibit relatively strong sensitivity to changes in bare ice albedo."

37. Line 527: contraction → reduction

**Responses:** Revised as suggested.

38. Line 545: Not sure that a change in temperature of 0.071°C is a noticeable temperature forcing

**Responses:** Thank you for your comment. We have replaced "noticeable" with "measurable" in the revised manuscript.

39. Line 582: exert a significant control.

**Responses:** Thank you for your comment. We have fixed it.

**Responses to reviewer #2's comments about the manuscript, entitled "Enhanced MODIS-derived ice physical properties within CoLM revealing bare ice-snow-albedo feedback over Greenland" (EGUSPHERE -2025-230)**

**General comments**

This paper has been improved by revisions, but corrections are still required.

**Responses:** We sincerely thank Reviewer #2 for the thoughtful and constructive feedback on our revised manuscript. We are encouraged to hear that the revisions have improved the quality of the paper. We fully acknowledge the remaining concerns raised and have carefully addressed each point in the revised manuscript. We greatly appreciate the reviewer's time, expertise, and effort in evaluating our work. A detailed point-by-point response is provided below, with specific changes clearly marked in red within this response document.

**Specific comments**

1. Line 39: There are remnants of ice age ice not in the GrIS, but this statement serves no purpose anyway and the sentence reads better without it.

**Responses:** We thank the reviewer for the suggestion. We have removed the statement as recommended to improve the clarity and conciseness of the sentence.

2. Line 47: What about basal drainage?

**Responses:** Thank you for the insightful comment. In our manuscript, the total mass loss from the GrIS is described as consisting of two main components: surface runoff and frontal ablation at glacier terminus (Cogley et al., 2011; Kochtitzky et al., 2023). Basal drainage refers to the subglacial transport of meltwater beneath the ice sheet (Cogley et al., 2011) and is an important hydrological process influencing glacier dynamics and sliding. In many studies, basal drainage is commonly included within meltwater runoff terms rather than quantified as a separate mass loss component.

Since basal drainage primarily concerns internal water routing beneath the glacier rather than the actual mass lost from the ice sheet to the ocean or atmosphere, it is not explicitly included in our current simplified mass loss categorization. However, we acknowledge that basal drainage plays a key role in modulating ice dynamics and meltwater export pathways.

We are happy to revise the manuscript to clarify this distinction or to explicitly mention basal drainage if the reviewer or editor considers it necessary.

**References:**

Cogley, J. G., Hock, R., Rasmussen, L. A., Arendt, A. A., Bauder, A., Braithwaite, R. J., Jansson, P., Kaser, G., Möller, M., Nicholson, L.: Glossary of glacier mass balance and related terms (p. 86), IHP-VII Technical Documents in Hydrology No, 2011.

Kochtitzky, W., Copland, L., King, M., Hugonnet, R., Jiskoot, H., Morlighem, M., Millan, R., Khan, S. A., and Noël, B.: Closing Greenland's mass balance: Frontal ablation of every Greenlandic glacier from 2000 to 2020, Geophys. Res. Lett., 50, e2023GL104095, https://doi. org/10.1029/2023GL104095, 2023.

3. Line 198: "relative error of 0.16%" in what?

**Responses:** Thank you for your comment. Thank you for the comment. We clarify that the "relative error of 0.16%" refers specifically to the relative error in bare-ice classification accuracy. This value comes from Antwerpen et al. (2022), who compared the threshold-based classification with Landsat 8 OLI observations and found a relative classification error of only 0.16%, supporting the robustness of the 0.6 threshold for bare-ice classification. We have revised the text accordingly for clarity.

4. Line 212: MCD43C3 is not "among" GLASS-AVHRR and C3S-v2 albedo products (which are not otherwise mentioned).

**Responses:** Thank you for your comment. We agree that "among" is not appropriate here since the other products are not mentioned elsewhere in the manuscript. We have revised the sentence accordingly to clarify the comparison: "Compared with the GLASS-AVHRR and C3S-v2 albedo products, MCD43C3 demonstrates superior performance for monitoring snow albedo, exhibiting the lowest bias and RMSE over snow and consistent performance across diverse snow cover conditions (Urraca et al., 2022)."

5. Line 230: Why not exclude all albedo values with low-quality indices? Why does quality also decrease at low SZA?

**Responses:** We thank the reviewer for pointing this out and apologize for the misstatement in the original sentence. In fact, we excluded all albedo values identified

with low-quality indices (4 and 5), regardless of SZA. The purpose of Figure 2b is to illustrate that low-quality albedo values predominantly occur at high SZA, and thus the exclusion of these values mainly affects retrievals under high-SZA conditions.

Regarding the reviewer's question about the decline in quality at low SZA, high SZA is the primary factor associated with reduced albedo retrieval quality, while low-quality indices at lower SZA can still arise due to factors such as persistent cloud contamination or unfavorable viewing geometries.

We have revised the sentence in the manuscript to clarify that all low-quality albedo values were excluded, and that the impact of this filtering is most significant for high-SZA conditions. The updated sentence now reads: "To ensure reliable satellite-retrieved bare ice physical properties, we excluded all albedo values identified with a low-quality index (4 or 5), regardless of the SZA. Figure 2b shows that the proportion of low-quality indices increases markedly when the SZA exceeds 70°, indicating that such filtering primarily affects high-SZA retrievals."

6. Line 245-247: "retrieved using the physical properties and SZA within the precomputed standalone SNICAR-Adv4 lookup table to match MCD43A3 band 2 BSA"

**Responses:** We thank the reviewer for the helpful suggestion to improve the clarity and flow of this sentence. The updated sentence in the manuscript now reads (revised portion marked in red in the response document): "Second, the bare ice physical properties (ice density and air bubble effective radius) are retrieved using the physical properties and SZA within the precomputed standalone SNICAR-Adv4 lookup table to match MCD43A3 band 2 BSA."

7. Line 261: Although the density-radius relationship is uncertain, it is important for this study. How have these values been selected?

**Responses:** We thank the reviewer for raising this important point. The density–bubble radius relationship for GrIS bare ice is indeed uncertain. In our study, we adopted a linear density-radius relationship as a first-order approximation following Whicker-Clarke et al. (2024). They point out that while the physical properties of bare ice can be constrained by SSA, the relationship between SSA, ice density, and air bubble radius is not unique. Importantly, there is an essentially one-to-one correspondence between SSA and albedo for a given solar zenith angle,

meaning that the specific combination of ice density and bubble size used to achieve a given SSA has little practical impact on the radiative transfer modeling.

Furthermore, we adopted 650 kg·m⁻³ as the lower bound for ice density based on recommendations from the SNICAR-ADv4 model developers (Whicker-Clarke et al., 2022). We acknowledge, as Reviewer #1 pointed out, that this value may be somewhat low compared to typical glacier ice densities. However, SNICAR-ADv4 achieves better agreement with observed spectral albedos when media with densities above approximately 650–700 kg·m⁻³ are classified and treated as ice layers rather than snow layers (Whicker-Clarke et al., 2022; Bender et al., 1997; Dadic et al., 2013). Thus, using this threshold helps improve the consistency between modeled and observed albedo properties in our simulations.

**References:**

Whicker-Clarke, A., Flanner, M. G., Dang, C., Zender, C. S., Cook, J. M., and Gardner, A. S.: SNICAR-ADv4: A physically based radiative transfer model to represent the spectral albedo of glacier ice, The Cryosphere, 16(4), 1197–1220, https://doi.org/10.5194/tc-16-1197-2022, 2022.

Bender, M., Sowers, T., and Brook, E.: Gases in ice cores, P. Natl. Acad. Sci., 94, 8343–8349, https://doi.org/10.1073/pnas.94.16.8343, 1997.

Dadic, R., Mullen, P. C., Schneebeli, M., Brandt, R. E., and Warren, S. G.: Effects of bubbles, cracks, and volcanic tephra on the spectral albedo of bare ice near the Transantarctic Mountains: Implications for sea glaciers on Snowball Earth, J. Geophys. Res.-Earth, 118, 1658–1676, https://doi.org/10.1002/jgrf.20098, 2013.

8. Line 272: This sentence seems meaningless when the SSA is determined by the specific ice density/bubble size combination.

**Responses:** We thank the reviewer for this comment. While it is true that SSA ultimately determines albedo, the statement emphasizes that among various combinations of ice density and bubble size that yield the same SSA, the modeled albedo remains effectively the same. This clarification is important to justify the use of a linear density-radius relationship as a first-order approximation in our study. It highlights that the precise combination of ice density and bubble size is less critical for albedo modeling than accurately capturing SSA itself, thereby supporting this method.

9. Line 308: Why is the average difference in Figure 4(i) not zero

**Responses:** We thank the reviewer for this insightful question. The average difference in Figure 4(i) is not exactly zero due to the stepwise nature of the equivalent black carbon (BC) concentration retrieval process. Specifically, in the standalone SNICAR-ADv4 model, we iteratively increased the input BC concentration in fixed increments of 20 ppb, starting from zero, until the simulated visible albedo dropped just below the MODIS-observed value. The corresponding BC concentration at that point was then recorded as the inferred equivalent BC value.

To reduce the systematic offset introduced by this approach, we subtracted half of the increment (10 ppb) as an empirical adjustment. Even with this correction, the retrieved BC concentration remains slightly higher than the value that would yield an exact albedo match, resulting in the small negative bias observed. While a finer step size could reduce this residual error, it would substantially increase the computational cost given the large number of pixels and time steps involved.

10. Line 315-326: This paragraph and Figure 5 would sit better before Figure 3.

**Responses:** Thank you for the helpful suggestion. We have moved Figure 5 ahead of the previous Figure 3, as recommended. To better highlight its relevance, we now describe Figure 5 in a standalone subsection titled 2.3 Parameter sensitivity of ice spectral albedo in SNICAR-ADv4. Following this section, and in accordance with Reviewer #1's suggestion, we have also added a synthetic description of the physical processes underlying bare ice metamorphism to bridge the controlled model behavior with natural conditions.

11. Line 334: The level of the ERA5 forcing data has to be stated.

**Responses:** We thank the reviewer for this important technical note. The ERA5 forcing data used in this study were obtained from the ECMWF's full-resolution, single-level (surface) dataset, with an hourly temporal resolution and $0.25° \times 0.25°$ spatial resolution. We also acknowledge that the original manuscript incorrectly described the spatiotemporal resolution of the ERA5 forcing data. The sentence in the Methods section has been revised accordingly to: "We conduct several offline CoLM simulations with the embedded SNICAR-ADv4 and SNICAR-AD schemes on a $0.25° \times 0.25°$ grid, driven by atmospheric forcing from the hourly single-level surface dataset of the European Centre for Medium-Range Weather Forecasts' fifth-generation reanalysis (ERA5) over the GrIS."

12. Line 359: Is shortwave albedo under direct radiation the same as BSA?

**Responses:** Thank you for the helpful comment. Yes, the shortwave albedo under direct radiation corresponds to the BSA, which represents the albedo under clear-sky conditions (i.e., direct-beam radiation or direct zenith illumination). We have clarified this terminology in the revised manuscript as follows: "In this study, shortwave albedo under direct radiation is treated as equivalent to the BSA, in accordance with the widely accepted terminology used in the MCD43C3 product."

13. Line 368: The reference to Figs 3c and e does not seem right.

**Responses:** Thank you for pointing this out. We have corrected the figure reference from "Figs. 3c and e" to "Figs. 4d and f" following the updated figure numbering in the revised manuscript.

14. Line 391: The y axis of Figure 6 (c) needs a label. The x and y axes would conventionally be the other way round.

**Responses:** Thank you for the helpful comment. In Figure 6c, the y-axis is labeled as "Mean exposed bare ice fraction", and the x-axis represents the relative percentage of grid cells. While we acknowledge that axis orientation can vary by convention, we chose a horizontal bar plot layout to maintain a compact and consistent arrangement across panels (a) to (c). We kindly ask for your understanding in maintaining the current design for visual clarity and layout cohesion.

[Figure]

Figure R1. Spatial distribution of (a) the fraction of land ice underlying the snowpack, excluding interior regions of the GrIS that remain fully snow-covered throughout JJA

(2000–2020), and (b) the mean exposed bare ice fraction during JJA over the same period, based on snow cover simulated by CoLM using the SNICAR-ADv4 scheme. Panel (c) shows the relative frequency distribution of mean exposed bare ice fraction, considering only grid cells with nonzero bare ice exposure. Each bar indicates the percentage of these grid cells whose mean bare ice fraction falls within a given interval, relative to the total number of bare ice grid cells.

15. Line 465: The application of the SNICAR-ADv4 has not itself reduced bias in albedo simulations; the calibration of SNICAR-Adv4 with albedo observations has.

**Responses:** We appreciate the opportunity to clarify the respective roles of model physics and observational calibration in improving albedo simulations. While SNICAR-ADv4 provides a more advanced representation of ice microstructure, including bubble-related scattering, the 38% reduction in modeled albedo bias is primarily attributed to the calibration against MODIS observations and the optimization of optical parameters. This improvement results from a two-step approach: first, implementing physically based model enhancements, and second, constraining key parameters using observational data. We have incorporated the relevant statements into the first paragraph of the Conclusions and Discussion section: "The application of SNICAR-ADv4, together with the integration of MODIS-derived bare ice properties, significantly improved albedo simulations by reducing the bias introduced by the default constant ice albedo treatment. Specifically, bias reductions of 38%, 50%, and 28% were achieved for broadband, visible, and near-infrared albedo, respectively. This improvement stems not only from the physically enhanced radiative transfer calculations over the ice column in SNICAR-ADv4, but also from the critical incorporation of MODIS-constrained ice optical properties, such as ice density and bubble radius. These additions provide better physical realism and representation of surface conditions across the ablation zone. "

16. Line 480: For a land model driven with ERA5 temperatures at a height above 2 m, these results do not suggest that the 2 m temperature increase contributes to snow melting; it is a response to the change in albedo and snow melting.

**Responses:** We sincerely appreciate the reviewer's insightful comment regarding the causality between temperature changes and snowmelt. We agree with the reviewer that the 2-m air temperature increase is not a cause of snowmelt but rather a response to the albedo reduction and associated surface energy balance changes. As the reviewer rightly notes, these temperature changes result from enhanced energy absorption due to reduced albedo, rather than being driven directly by the

ERA5-prescribed atmospheric forcing. To avoid confusion, we have removed the original misstatement, as the role of snowmelt-albedo feedback, in which melting exposes darker underlying surfaces and further amplifies warming, is already described later in the text.

**Minor comments:**

1. Line 22: "model to represent"

**Responses:** Thank you for your comment. We have corrected it.

2. Line 33: "expected in"

**Responses:** Thank you for your comment. We have corrected it.

3. Line 53: "discharge"is not the right word here.

**Responses:** Thank you for pointing this out. We have replaced "ice discharge" with "ice melt" in the revised manuscript.

4. Line 56: "the regional"

**Responses:** Thank you for your comment. We have corrected it.

5. Line 113: "prescribed as constant values"

**Responses:** Thank you for your comment. We have corrected.

6. Line 119: Delete "(ESMs)"

**Responses:** Thank you for your comment. We have removed "(ESMs)" as suggested.

7. Line 120:"retrieved from"

**Responses:** Thank you for your comment. We have corrected.

8. Line 122:"ESM"

**Responses:** Thank you for your attention to this detail. We confirm that "ELM" (E3SM Land Model) is correct here as it specifically refers to the land component of E3SM. To avoid confusion, we have added a clarifying phrase in the sentence introducing ELM earlier in the revised manuscript: "To advance ice radiative transfer modeling in Earth system models, Whicker-Clarke et al. (2024) incorporated SNICAR-ADv4 into the E3SM (specifically its land component, ELM), in which the GrIS ice physical properties are retrieved from the satellite observation data."

9.  Line 137: "exposed by snow melting during the ablation season"

**Responses:** Thank you for your comment. We have corrected it.

10.  Line146: "quantifies"

**Responses:** Thank you for your comment. We have corrected it.

11.  Line 161: "grain size and shape"

**Responses:** Thank you for your comment. We have corrected it.

12.  Line 207:"evaluating"

**Responses:** Thank you for your comment. We believe the original usage of "evaluate" is grammatically correct in this context, as it refers to the second of two purposes for using the MODIS MCD43C3 product. Nonetheless, to improve clarity and avoid any ambiguity, we have revised the sentence as follows: "The MODIS MCD43C3 product (Schaaf et al., 2002) is used to retrieve bare ice physical properties through standalone SNICAR-ADv4, and to evaluate the CoLM-simulated albedo over the GrIS bare ice regions."

13.  Line 261: "correspond"

**Responses:** Thank you for your comment. We have corrected it.

14.  Line 269: "(1)"

**Responses:** Thank you for pointing this out. We have revised the notation from "(Eq. 1)" to "(1)" in accordance with the journal's formatting style.

15.  Line 270: "(2)"

**Responses:** Thank you for pointing this out. We have revised the notation from "(Eq. 2)" to "(2)" in accordance with the journal's formatting style.

16.  Line 277: "note, however, that"

**Responses:** Thank you for your comment. We have corrected it.

17.  Line 289: "3d and e"

**Responses:** Thank you for pointing this out. We have corrected the reference to "4d and e" in the revised manuscript, reflecting the updated figure order.

18.  Line 322: "density and air bubble"

**Responses:** Thank you for your comment. We have corrected it.

19. Line 342: Delete "participated"

**Responses:** Thank you for your suggestion. We have removed the word "participated" as recommended.

20. Line 364: "c-f"

**Responses:** Thank you for your comment. We have corrected it.

21. Line 367: "shows"

**Responses:** Thank you for your comment. We have corrected it.

22. Line 369: "represents"

**Responses:** Thank you for your comment. We have corrected it.

23. Line 371: "high values in the area"

**Responses:** Thank you for your comment. We have corrected it.

24. Line 376: "4g"

**Responses:** Thank you for pointing this out. We have corrected the reference to "5g" in the revised manuscript, reflecting the updated figure order.

25. Line 393: "for JJA"

**Responses:** Thank you for your comment. We have removed the redundant definite article "the" before "JJA" throughout the manuscript for consistency. The corrected phrase now reads 'for JJA' as suggested.

26. Line 435: "is consistent"

**Responses:** Thank you for your comment. We have corrected it.

27. Line 436: "presents"

**Responses:** Thank you for your comment. We have corrected it.

28. Line 464: "property changes"

**Responses:** Thank you for your comment. We have corrected it.

29. Line 500: "in changes"

**Responses:** Thank you for your comment. We agree and have revised the phrase to "in changes" to improve clarity and conciseness.

30. Line 502: "which has a"

**Responses:** Thank you for your comment. We have corrected it.

31. Line 517: Unites of snow water equivalent are required.

**Responses:** Thank you for your comment. We have added the unit "mm" for snow water equivalent at the relevant location in the manuscript.

32. Line 533: "made enhanced"

**Responses:** Thank you for your comment. We have revised the phrase "made enhanced" to improve grammatical clarity. It now reads: "...and made enhanced MODIS-informed bare ice physical properties..."

33. Line 535: "to ice metamorphism"

**Responses:** Thank you for your comment. We have corrected it.

34. Line 538: "SNICAR-ADv4 and SNICAR-AD schemes"

**Responses:** Thank you for your comment. We have corrected it.

35. Line 545: "in the ablation region" or "in ablation regions"

**Responses:** Thank you for your comment. We have corrected it.

36. Line 562: "physical"

**Responses:** Thank you for your comment. We have corrected it.